EMBO
Molecular Medicine

# Adaptive RSK-EphA2-GPRC5A signaling switch triggers chemotherapy resistance in ovarian cancer

Lidia Moyano-Galceran[1] [ID], Elina A Pietilä[2,†] [ID], S Pauliina Turunen[1,†], Sara Corvigno[3,4], Elisabet Hjerpe[5], Daria Bulanova[6], Ulrika Joneborg[7], Twana Alkasalias[1,8], Yuichiro Miki[1,9], Masakazu Yashiro[9], Anastasiya Chernenko[2], Joonas Jukonen[2], Madhurendra Singh[1], Hanna Dahlstrand[3,4], Joseph W Carlson[3] & Kaisa Lehti[1,2,*] [ID]

## Abstract

Metastatic cancers commonly activate adaptive chemotherapy resistance, attributed to both microenvironment-dependent phenotypic plasticity and genetic characteristics of cancer cells. However, the contribution of chemotherapy itself to the non-genetic resistance mechanisms was long neglected. Using high-grade serous ovarian cancer (HGSC) patient material and cell lines, we describe here an unexpectedly robust cisplatin and carboplatin chemotherapy-induced ERK1/2-RSK1/2-EphA2-GPRC5A signaling switch associated with cancer cell intrinsic and acquired chemoresistance. Mechanistically, pharmacological inhibition or knockdown of RSK1/2 prevented oncogenic EphA2-S897 phosphorylation and EphA2-GPRC5A co-regulation, thereby facilitating a signaling shift to the canonical tumor-suppressive tyrosine phosphorylation and consequent downregulation of EphA2. In combination with platinum, RSK inhibitors effectively sensitized even the most platinum-resistant EphA2[high], GPRC5A[high] cells to the therapy-induced apoptosis. In HGSC patient tumors, this orphan receptor GPRC5A was expressed exclusively in cancer cells and associated with chemotherapy resistance and poor survival. Our results reveal a kinase signaling pathway uniquely activated by platinum to elicit adaptive resistance. They further identify GPRC5A as a marker for abysmal HGSC outcome and putative vulnerability of the chemoresistant cells to RSK1/2-EphA2-pS897 pathway inhibition.

**Keywords** chemotherapy; EphA2; GPRC5A; HGSC; resistance
**Subject Categories** Cancer; Signal Transduction

## Introduction

Despite advances in anti-cancer treatments, majority of patients with disseminated metastases eventually recur with an increasingly therapy-resistant disease (Dagogo-Jack & Shaw, 2018). Both intrinsic and acquired drug resistance mechanisms contribute to tumor heterogeneity and evolution of genetically resistant cancer clones (McGranahan & Swanton, 2017; Dagogo-Jack & Shaw, 2018). Extensive evidence also indicates that tumor microenvironment (TME)-dependent phenotypic plasticity contributes to the therapy resistance and recurrent growth (Fischer *et al*, 2015; Zheng *et al*, 2015; Senthebane *et al*, 2017). Although chemotherapy-induced changes in both the cancer cells and the TME have been linked to tumor aggressiveness (Norouzi *et al*, 2018; Redfern *et al*, 2018), the effects of chemotherapy itself on the non-genetic, adaptive signaling mechanisms activated in the treatment-escaping cancer cells remain elusive.

To dynamically communicate within the TME, tumor cells utilize cell surface receptors (Friedl & Alexander, 2011). The erythropoietin-producing hepatocellular receptor A2 (EphA2) is a widely expressed member of the largest receptor tyrosine kinase (RTK) family, the Eph receptors. EphA2 signals in a context-dependent and dual manner: (i) via ephrinA ligand-induced auto-phosphorylation at the cytoplasmic tyrosine residues, which can occur in connection with epithelial cell adhesion, and generally inhibits oncogenic signaling; or (ii) by ligand-independent signaling, whereby EphA2 is phosphorylated at cytoplasmic S897 residue, driving downstream pro-tumorigenic signaling upon crosstalk with other RTKs and signaling molecules (Gucciardo *et al*, 2014; Riedl & Pasquale, 2015; Kania & Klein, 2016; Zhou & Sakurai, 2017).

1 Department of Microbiology, Tumor and Cell Biology, Karolinska Institutet, Stockholm, Sweden
2 Research Programs Unit, Individualized Drug Therapy, University of Helsinki and Helsinki University Hospital, Helsinki, Finland
3 Department of Oncology and Pathology, Karolinska Institutet, Stockholm, Sweden
4 Department of Immunology, Genetics and Pathology, Uppsala University, Uppsala, Sweden
5 Department of Obstetrics and Gynecology, Visby Hospital, Visby, Sweden
6 Institute for Molecular Medicine Finland, FIMM, University of Helsinki, Helsinki, Finland
7 Division of Pelvic Cancer, Department of Women's and Children's Health, Karolinska Institutet and University Hospital, Stockholm, Sweden
8 Research Centre, Salahaddin University-Erbil, Erbil, Iraq
9 Department of Gastroenterological Surgery, Osaka City University Graduate School of Medicine, Osaka, Japan
*Corresponding author. Tel: +46 8 524 852 54; E-mail: kaisa.lehti@ki.se
†These authors contributed equally to this work

Reportedly, the kinases Akt, PKA, and p90 ribosomal S6 kinases (RSK/p90-RSK/S6KA) can mediate tumor-promoting EphA2-S897 phosphorylation (Miao et al, 2009; Zhou et al, 2015; Barquilla et al, 2016). Among the RSK family, RSK1 and RSK2 support tumor growth and survival, whereas RSK3 and RSK4 are frequently downregulated in aggressive cancers (Casalvieri et al, 2017). In the context-dependent regulation, EphA2-pS897 signaling has been linked to over-activation of EphA2, Src-kinase activation, and EphA2 cleavage by matrix metalloproteinase MMP14/MT1-MMP (Sugiyama et al, 2013; Koshikawa et al, 2015; Hamaoka et al, 2018). Through such oncogenic signaling crosstalk, EphA2 can alter cell–cell contacts and extracellular matrix (ECM) adhesion or degradation to promote anchorage-independence, invasion in collagenrich TME, drug resistance, and stem-like properties (Thaker et al, 2004; Lu et al, 2008; Sugiyama et al, 2013; Zhou & Sakurai, 2017; Giorgio et al, 2018).

Ovarian cancer (OC) is the most lethal gynecologic malignancy (Siegel et al, 2018). High-grade serous ovarian cancer (HGSC) accounts for approximately 70% of diagnosed cases, majority of which are in metastatic stages (Seidman et al, 2004; Kobel et al, 2010; Torre et al, 2018). Metastatic HGSC is associated with aggressive dissemination in the abdominal cavity, which occurs via detachment of OC cells from the primary tumor to the peritoneal fluid, followed by accumulation of the metastatic cells and multicellular aggregates in ascites (Kenny et al, 2007; Hjerpe et al, 2018). Upon exposure to specific cues, OC cells adhere and grow as solid metastatic lesions in peritoneal organs, including the fatty omentum as the preferred site for invasion and induction of collagen-rich desmoplastic TME (Kenny et al, 2007, 2011; Luo et al, 2016).

The relatively effective first-line therapy for HGSC patients is debulking surgery coupled to platinum-based chemotherapy (Marchetti et al, 2010). Despite initial treatment response, most HGSCs recur, often as a repeatedly chemo-sensitive disease (Pfisterer & Ledermann, 2006; Armbruster et al, 2018). This indicates that besides genetic changes and selection, more plastic resistance mechanisms are activated upon the aggressive disease progression (Friedl & Alexander, 2011). Targeting these mechanisms could provide effective combinatorial treatments urgently needed to eliminate also the chemotherapy-escaping OC (micro)metastases from sustaining aggressive tumor evolution.

Frequently overexpressed in OC, EphA2 associates with high tumor grade, advanced stage, and poor clinical outcome (Thaker et al, 2004). It has been recognized as a putative target to block HGSC progression, although currently developed molecular-targeted therapies lack proof for specificity and efficacy (Landen et al, 2005b; Petty et al, 2018). In adhesion-dependent signaling, EphA2 cooperates with integrins, the transmembrane receptors that link the ECM to cell cytoskeleton (Hamidi & Ivaska, 2018). Moreover, the G-protein coupled receptor Class C, Group 5, Member A (GPRC5A) has been identified as an interactor of EphA2 and $\beta1$-integrin (Bulanova et al, 2017). While tumor-suppressive and oncogenic functions have been reported for this orphan receptor, possible GPRC5A functions in OC remain unknown (Zhou & Rigoutsos, 2014).

Intrigued by our unexpected observation of platinum-induced EphA2 upregulation in ex vivo 3D collagen cultures of HGSC patient cells, we used relevant cell models and clinical tumor material to understand the EphA2-GPRC5A pathway and its clinical implications in OC. Our results uncover a robust platinum-induced switch in EphA2 signaling duality via RSK activation, which pharmacological reversal allowed elimination of the otherwise resistant GPRC5A overexpressing cells.

# Results

## Cisplatin treatment leads to EphA2 upregulation in patient-derived HGSC cells ex vivo

For investigating HGSC signaling and TME-dependent resistance to platinum chemotherapy, we established ex vivo cultures from the ascites of treatment-naïve patients with metastatic disease (Table 1). The fresh patient cells were plated to ascites-like culture growing spontaneously as suspension cells and spheres, or embedded in 3D collagen, which typifies the collagen-rich desmoplastic microenvironment around solid HGSC metastatic lesions (Kenny et al, 2007). By immunofluorescence, these cells were 60–90% positive for the nuclear HGSC marker PAX8 (Fig EV1A; Laury et al, 2010). Ex vivo cell responses to cisplatin were variable with part of the patient cultures showing treatment resistance particularly when embedded in collagen (Fig EV1B). In such culture, cisplatin affected the cell viability by increased apoptosis (Fig EV1C–E, cleaved caspase-3).

The cells grew in 3D collagen as colonies positive for cytokeratin 7 (CK7; epithelial HGSC marker; Lengyel, 2010), with or without surrounding residual CK7$^-$, vimentin$^+$ mesenchymal cells (Fig 1A; see OCKI_p01 and OCKI_p02, respectively). The CK7$^+$ cell morphology ranged from compact sphere-forming cells, prominent in cultures OCKI_p01 and OCKI_p02, to round cells in looser grape-like colonies in relatively resistant cultures OCKI_p03 and OCKI_p06 (Fig 1A). Considering the rounded collagen invasive phenotype of OCKI_p03 and OCKI_p06 cells, resembling the reported EphA2-dependent breast cancer cell phenotypes (Sugiyama et al, 2013), and EphA2 association with OC clinical outcome (Thaker et al, 2004), we analyzed EphA2 in these ex vivo cultures by immunofluorescence. Significantly, cisplatin treatment led to over twofold increased EphA2 intensity in the treatment-escaping OCKI_p01, OCKI_p03, and OCKI_p06, while OCKI_p02 cells were positive for EphA2 also prior treatment (Fig 1B and C; OCKI_p01: $3.8 \pm 0.2$, OCKI_p03: $3.5 \pm 0.2$, and OCKI_p06: $2.0 \pm 0.1$-fold increase, $P \leq 0.008$).

## Platinum induces an oncogenic feedback response via EphA2 tyrosine–serine phosphorylation switch in OC cell lines and patient-derived cells

The context-dependent EphA2 signaling can occur via ligand-induced tyrosine auto-phosphorylation, generally considered tumor-suppressive, or via oncogenic, ligand-independent phosphorylation of the S897 residue (Gucciardo et al, 2014; Zhou & Sakurai, 2017). To examine whether platinum chemotherapy affects this EphA2 signaling duality, OVCAR3, OVCAR4, and OVCAR8 cells were first treated with up to 20 μM cisplatin for 72 h (see Appendix Fig S1A and B for cell characterization). In all these human OC cell lines, EphA2 was constitutively expressed and increased after platinum treatment (Fig 2A). The ligand-independent EphA2-pS897 was likewise enhanced. The tumor-suppressive EphA2-pY588 was increased

**Table 1. Patient information.**

| Patient | Origin[A] | Stage | Residual tumor size | BRCA status | Platinum–taxane regimen | Response | Follow-up |
|---|---|---|---|---|---|---|---|
| OCKI_p01 | HGS-O | IVB | 0 mm | Mut | Yes | CR | NED |
| OCKI_p02 | HGS-O | IIIC | 0 mm | WT | Yes | CR | PD |
| OCKI_p03 | HGS-O | IVB | > 2 cm | Mut | Yes | PR | PD |
| OCKI_p04 | HGS-FP | IIB | 0 mm | WT | Yes | CR | NED |
| OCKI_p06 | HGS-FP | IIIC | 0 mm | WT | Yes | CR | PD |
| OCKI_p10 | HGS-FP | IIIC | 0 mm | WT | Yes | CR | NED |
| OCKI_p11 | HGS-O | IVB | > 2 cm | WT | Yes | PR | PD |
| OCKI_p13 | HGS-FP | IIIC | 0 mm | WT | Yes | CR | NED |
| OCKI_p20 | HGS-FP | IIIC | 0 mm | Mut | Yes | CR | NED |
| OCKI_p22 | HGS-FP | IIIC | < 5 mm | WT | Yes | CR | NED |
| OCKI_p25 | HGS-FP | IIIB | < 1 cm | WT | Yes | PD | PD |
| OCKI_p27 | HGS-FP | IIIC | 0 mm | NA | NA | NA | NA |
| OCKI_p28 | CCC | IIIC | > 2 cm | NA | Yes | NA | NA |

HGSC origin[A]: O, ovary; FP, fallopian tube. *Abbreviations*: CCC, clear cell carcinoma; WT, wild type; Mut, mutant; CR, complete response; PR, partial response; PD, progressive disease; NED, no evidence of disease; NA, no available data.

in OVCAR3 and to a less extent in OVCAR4, but was low in OVCAR8 with and without cisplatin (Fig 2A–C). Notably, the pS897/pY588 ratio increased in all three cell models compared to untreated controls (Fig 2B; OVCAR3: 5 μM cisplatin 2.7 ± 0.6, OVCAR4: 10 μM cisplatin 4.2 ± 0.5, and OVCAR: 20 μM cisplatin 3.5 ± 0.1-fold increase, $P \leq 0.047$). Moreover, OVCAR3 and OVCAR4 with low pS897/pY588 ratio prior treatment were sensitive to cisplatin, whereas platinum-resistant OVCAR8 had constitutive oncogenic EphA2-pS897 dominance (Fig 2C and D; cell viability at 20 μM cisplatin: OVCAR3 24.3 ± 10.7% and OVCAR4 20.9 ± 6.8% vs. OVCAR8 84.7 ± 3.3%, $P < 0.001$).

Patient-derived HGSC cultures likewise expressed EphA2, which was increased after cisplatin treatment (Fig 2E, Appendix Fig S2A, see Appendix Fig S2B and C for PAX8 positivity and mutant TP53 pattern of nutlin unresponsiveness). Coincidentally, oncogenic EphA2-pS897 increased in all six patient cells, whereas EphA2 auto-phosphorylation showed an opposite pattern with EphA2-pY588 in the untreated cells declining progressively after treatment with increasing concentrations of cisplatin (Fig 2E, Appendix Fig S2A). Thus, the pS897/pY588 ratio was significantly increased in these patient-derived cells (Fig 2E and F; 5 μM cisplatin 3.0 ± 1.3, 10 μM cisplatin 4.5 ± 1.7, 20 μM cisplatin 6.1 ± 1.3-fold increase, $P \leq 0.02$).

The current suggested platinum chemotherapy for OC patients is carboplatin, a non-inferior but less toxic platinum-derivate that also causes less unspecific apoptosis *in vitro* (du Bois *et al*, 2003; Good-isman *et al*, 2006). To validate the specific effect of platinum on EphA2, OVCAR4 and OVCAR8 were treated with up to 80 μM carbo-platin (higher concentrations than cisplatin due to lower chemical reactivity; Alberts & Dorr, 1998). After treatment, EphA2 (total and pS897) was enhanced and the tumor-suppressive EphA2-pY588 decreased in OVCAR4 (Fig 2G). As a result, carboplatin significantly increased the pS897/pY588 ratio (Fig 2G and H; 3.0 ± 0.2-fold at 80 μM carboplatin, $P = 0.014$), whereas the platinum-resistant OVCAR8 had high pS897/pY588 prior and after treatment (Fig 2G and H; 2.1 ± 0.1-fold higher in untreated OVCAR8 than

corresponding OVCAR4, $P = 0.048$). Altogether, these results reveal a previously unappreciated induction of a robust oncogenic EphA2 phosphorylation switch by platinum chemotherapy in HGSC cells.

### Platinum triggers an oncogenic EphA2-S897 phosphorylation *in vivo*

To define the effect of platinum in EphA2 signaling *in vivo*, OVCAR4 cells were lentivirally transduced to express *Renilla* luciferase and injected intraperitoneally in severe combined immunodeficient (SCID) female mice. All mice developed tumors in the abdominal cavity (Figs 3A and EV2A). These tumors grew as widely disseminated foci in the omentum and other peritoneal organs, coincident with the accumulation of ascites, thus mimicking HGSC dissemination in patients (Fig EV2A; Kenny *et al*, 2011). Carboplatin effectively reduced the tumor burden and eliminated the ascites (Fig 3B and C; $P \leq 0.01$). In the solid omental and peritoneal tumors, carbo-platin had negligible effects on proliferation (Ki67), whereas apopto-sis (TUNEL and clCasp3) was increased (Figs 3H and I, and EV2B–D; $P \leq 0.042$). However, residual tumor foci remained, modeling the resistant metastatic lesions with potential for aggressive disease progression in patients (Fig 3D; Pfisterer & Ledermann, 2006; Armbruster *et al*, 2018). By immunofluorescence, total EphA2 and the oncogenic EphA2-pS897 were enhanced in the residual carbo-platin-treated tumors as compared to untreated controls (Fig 3E–H; EphA2 1.4 ± 0.4, EphA2-pS897 1.6 ± 0.1-fold increase; $P \leq 0.009$). In the carboplatin-treated tumors, EphA2-pS897 and clCasp3 local-ized to different tumor cells and areas (Fig 3H). These results suggest that the treatment-escaping HGSC cells activated oncogenic EphA2 signaling in response to platinum chemotherapy also *in vivo*.

### EphA2 phosphorylation switch is associated with platinum resistance

To address the possible relationship between EphA2 signaling and platinum resistance, we used the human HGSC model of TYK-nu

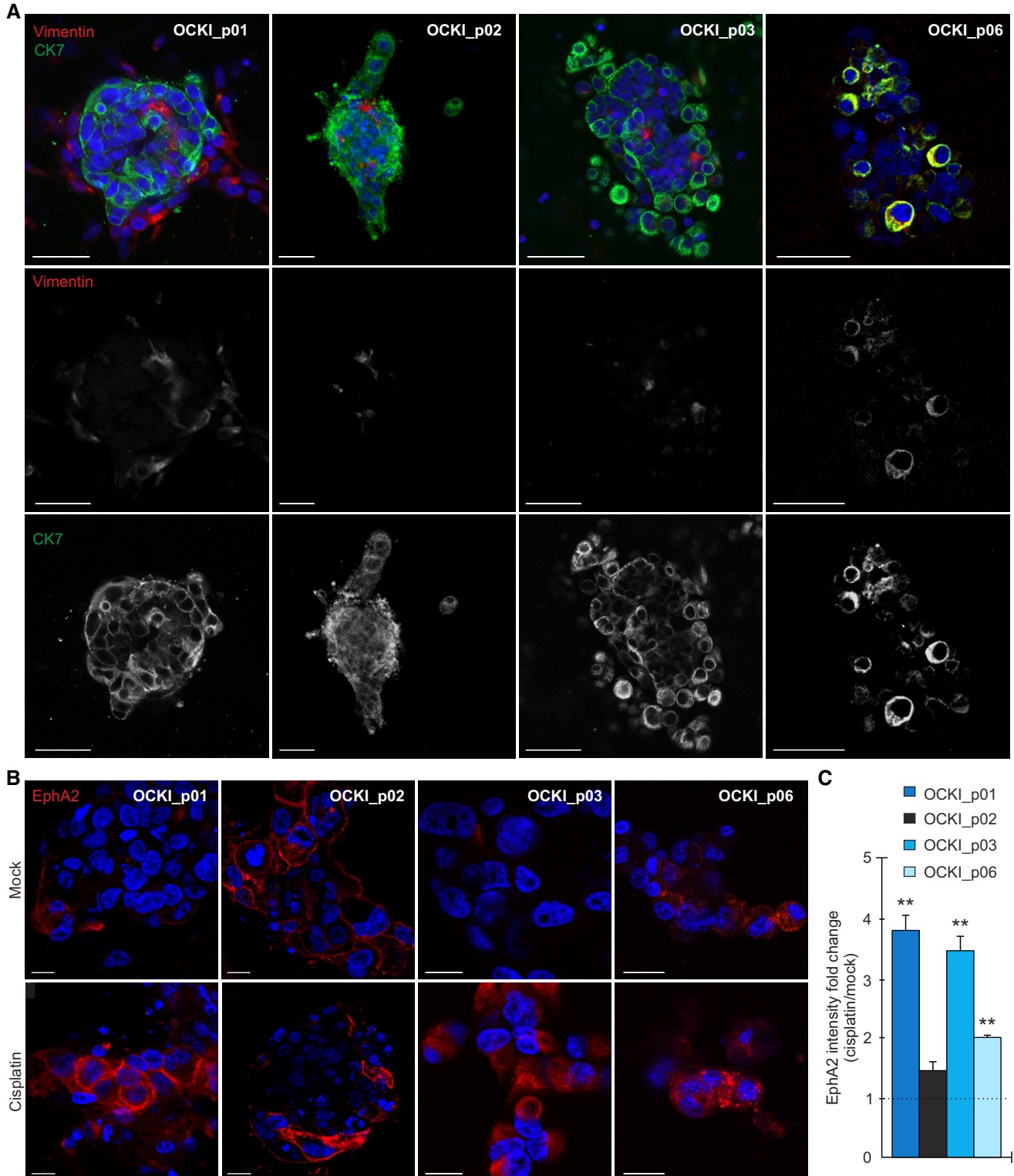

**Figure 1. Cisplatin treatment leads to EphA2 upregulation in *ex vivo* HGSC cultures.**

A Confocal micrographs show cytokeratin 7 (CK7, green) and vimentin (red) in patient-derived HGSC cells cultured in 3D collagen for 7 days. Scale bars: 50 μm.

B Confocal micrographs of EphA2 (red) in HGSC cells cultured in 3D collagen for 7 days and treated for 72 h with 10 μM cisplatin (5 μM for OCKI_p01). The intensity of EphA2 is comparable only between mock and treatment conditions for each patient. Scale bars: 20 μm.

C Chart illustrates EphA2 fold change after treatment. Mock is set to one. N = 3.

Data information: In (C), data are presented as mean fold change (SD). **P < 0.01. Exact P-values are provided in Appendix Table S10, Student's t-test.

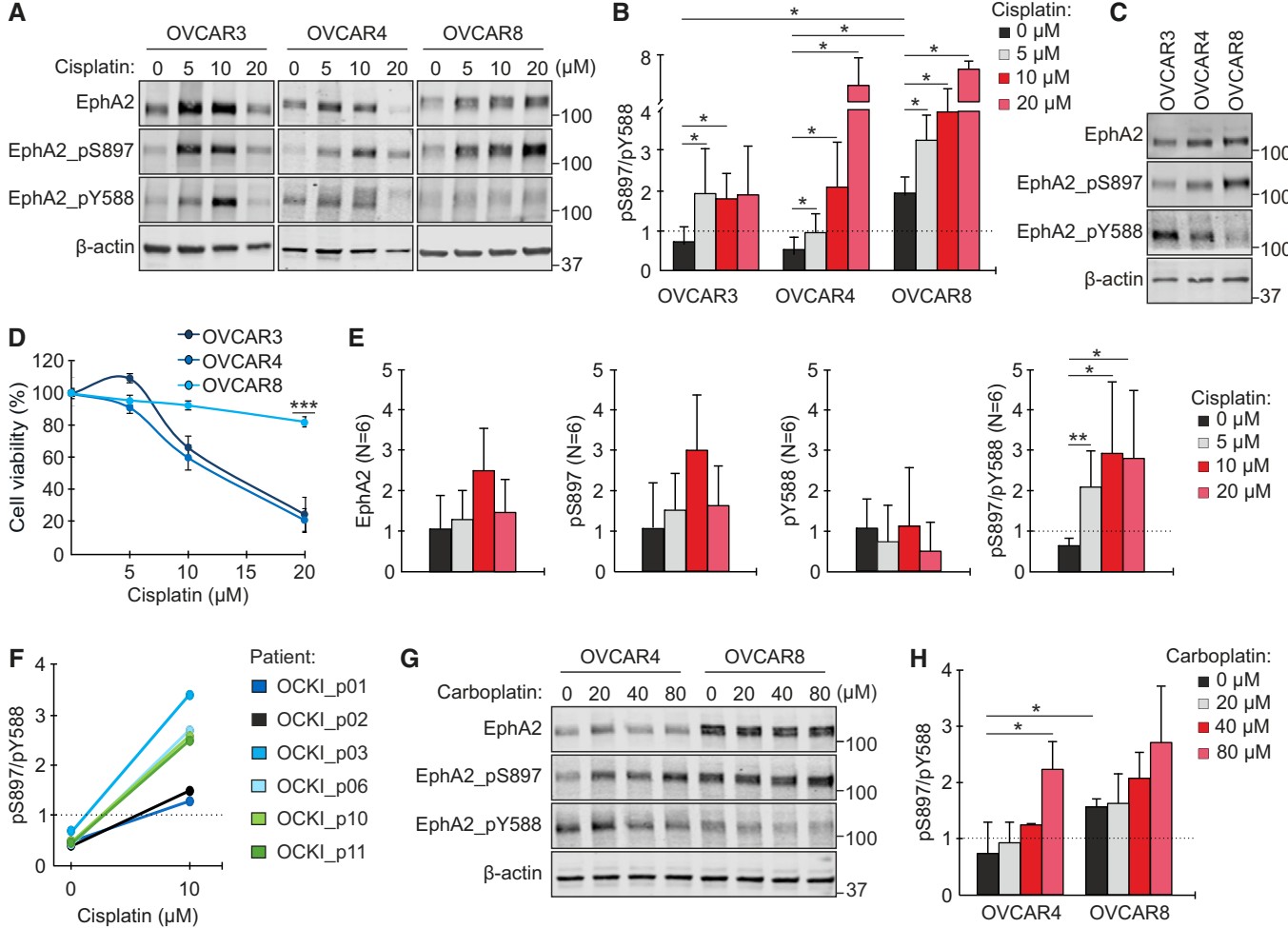

**Figure 2. OC cell lines and patient-derived cells undergo an EphA2 phosphorylation switch upon platinum treatment.**

A, B  EphA2 (total and phosphorylated at S897 or Y588) in OVCAR3, OVCAR4, and OVCAR8 after treatment with 0–20 μM cisplatin for 72 h was assessed by immunoblotting (A) and quantified for pS897/pY588 ratio (B). N = 4.

C  EphA2 (total and phosphorylated) in corresponding untreated cells. The same β-actin detection for these samples is shown in Appendix Fig S1B.

D  Cytotoxicity assay results after cell treatment with 0–20 μM cisplatin for 72 h. N = 6.

E  Quantitative assessment of EphA2 (total and phosphorylated) and pS897/pY588 ratio in early passage patient-derived HGSC cultures treated with 0–20 μM cisplatin for 72 h (see immunoblots in Appendix Fig S2A). N = 6 patients, pooled.

F  Corresponding EphA2 pS897/pY588 ratios for individual patient cells.

G, H  EphA2 (total and phosphorylated) in OVCAR4 and OVCAR8 after treatment with 0–80 μM carboplatin for 72 h (G) along with pS897/pY588 quantification (H). N = 3.

Data information: In (B, D–E, and H), data are presented as mean (SD). *$P < 0.05$, **$P < 0.01$, ***$P < 0.001$. Exact $P$-values are provided in Appendix Table S10, Student's $t$-test.

Source data are available online for this figure.

cells and their resistant subline TYK-nu.R, originally generated by continuous exposure to cisplatin (Yoshiya *et al*, 1989; Domcke *et al*, 2013). In line with our OVCAR and patient cell results, platinum treatment of TYK-nu enhanced EphA2 (total and pS897) and reduced the tumor-suppressive EphA2-pY588 (Fig 4A), thus increasing pS897/pY588 ratio (Fig 4B; 5 μM cisplatin 8.6 ± 0.3, 10 μM cisplatin 8.0 ± 0.4-fold increase, $P \leq 0.009$). Moreover, TYK-nu.R had 2.8 ± 0.2-fold higher pS897/pY588 than TYK-nu prior treatment (Fig 4A–D; $P = 0.004$), in a similar manner as with the platinum-resistant OVCAR8 relative to the more sensitive OVCAR4. Cisplatin further increased EphA2 pS897/pY588 also in TYK-nu.R (Fig 4A and B; 10 μM cisplatin 2.6 ± 0.4-fold increase, $P \leq 0.033$).

Importantly, carboplatin triggered an analogous oncogenic switch via coincidently increased EphA2-pS897 and decreased EphA2-pY588 in TYK-nu, while in TYK-nu.R, pS897/pY588 was high with and without carboplatin (Fig 4C). Therefore, the EphA2 signaling switch occurs as a HGSC cell response to platinum chemotherapy and is associated with increased treatment resistance among both TYK-nu and OVCAR cells.

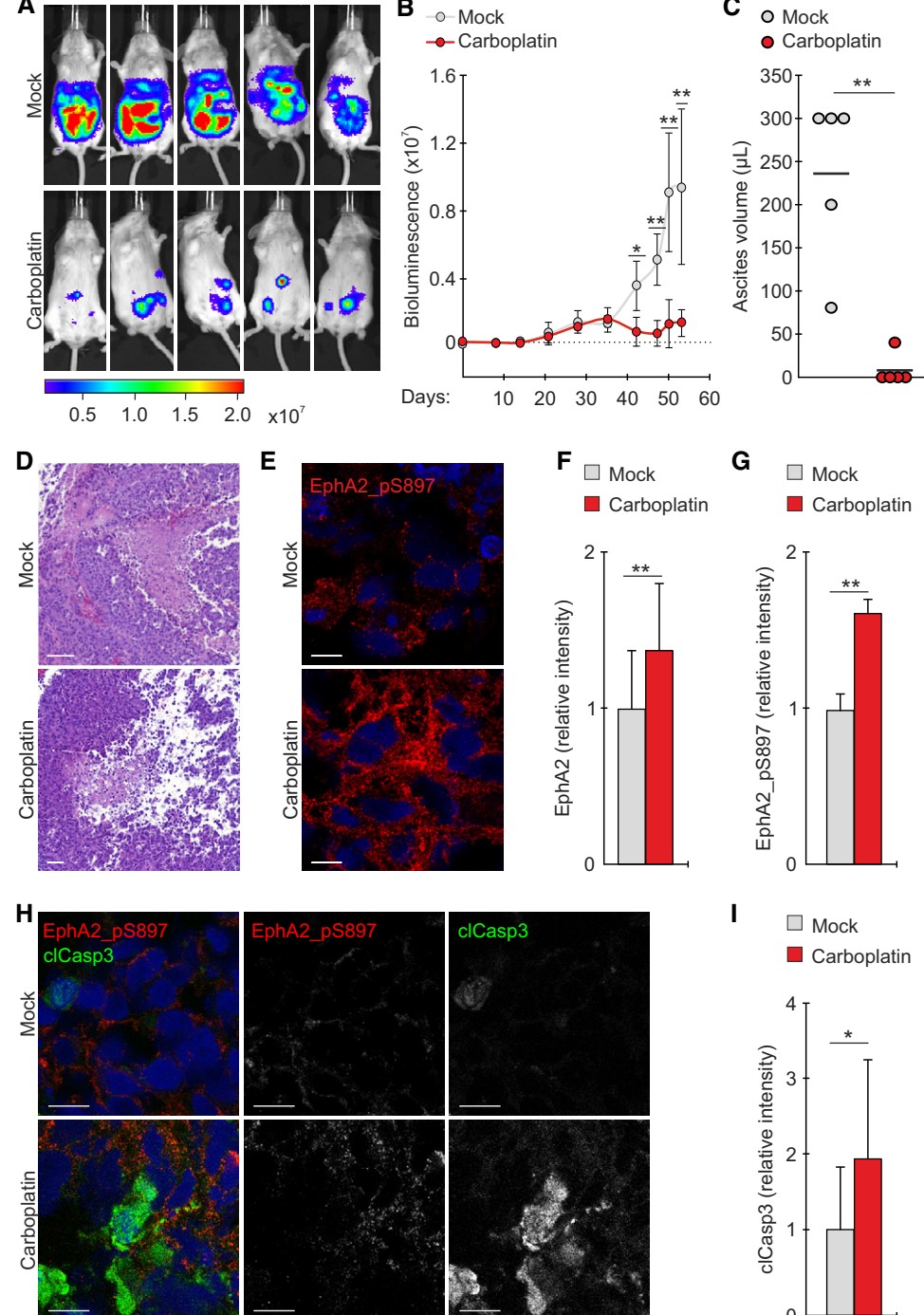

**Figure 3. Platinum treatment promotes ligand-independent oncogenic EphA2 phosphorylation *in vivo*.**

A    Bioluminescence images visualize mock- and carboplatin-treated OVCAR4 xenograft tumors (day 53; after 13-d i.p. treatment). Color scale unit: p/sec/cm²/sr.

B, C    Charts illustrate the tumor growth (B) and ascites volumes at the end of the follow-up (C). Carboplatin treatment started on day 35. Bioluminescent unit: p/sec/cm²/sr. *N* = 5 mice/group.

D, E    Representative micrographs of hematoxylin–eosin staining (D) and EphA2-pS897 immunofluorescence (E) in the xenografts tumors. Scale bars: 100 μm (D), 10 μm (E).

F, G    Quantitative assessment of tumor EphA2 (F) and EphA2-pS897 (G). *N* = 5 mice/group.

H, I    Confocal micrographs of tumor EphA2-pS897 (red) and cleaved caspase-3 (H; clCasp3, green) and corresponding clCasp3 quantification (I). Scale bars: 20 μm. *N* = 5 mice/group.

Data information: In (B and C; F and G; and I), data are presented as mean (SD). **P* < 0.05, ***P* < 0.01. Exact *P*-values are provided in Appendix Table S10, Mann–Whitney *U*-test.

Source data are available online for this figure.

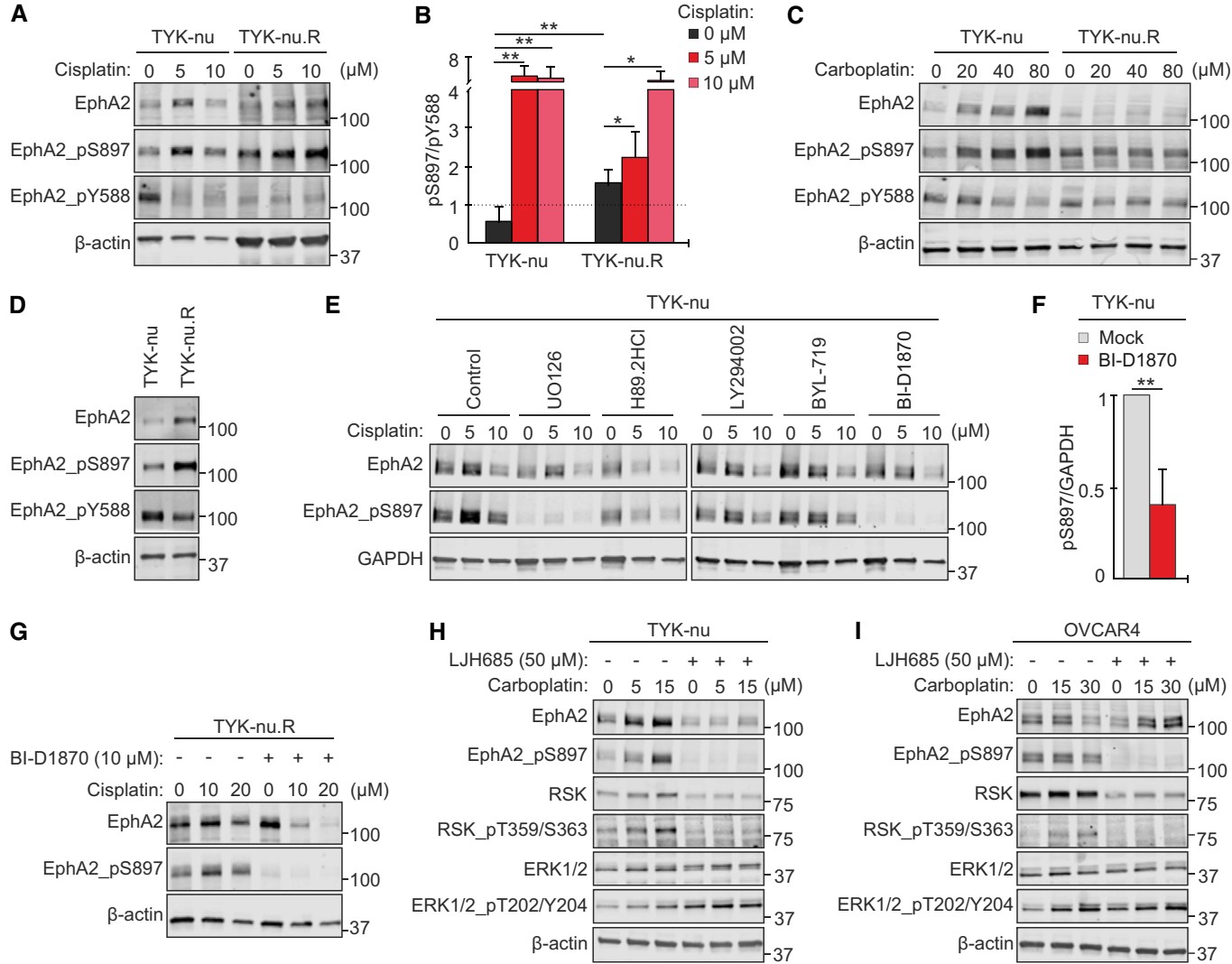

**Figure 4. Oncogenic EphA2 signaling switch is associated with platinum resistance and mediated by treatment-activated ERK1/2-RSK axis in OC cells.**

A, B   Immunoblot images show EphA2 (A; total, pS897, and pY588) and corresponding pS897/pY588 ratios (B) of TYK-nu and TYK-nu.R after treatment with 0–10 μM cisplatin for 72 h. N = 5.

C, D   EphA2 (total and phosphorylated) in the cells treated (C) and untreated (D) with 0–80 μM carboplatin for 72 h.

E, F   EphA2 (total and pS897) in TYK-nu after treatment with inhibitors against MEK (UO126, 10 μM), PKA (H89.2HCl, 10 μM), PI3K (LY294002, 10 μM) or its alpha subunit (BYL-719, 5 μM), and RSK (BI-D1870, 10 μM) in combination with 0–10 μM cisplatin for 72 h (E). Corresponding EphA2-pS897 quantification for BI-D1870 treatment (F). N = 5.

G   EphA2 (total and pS897) in TYK-nu.R treated with 0–20 μM cisplatin without or with 10 μM BI-D1870 for 72 h.

H, I   EphA2, RSK, and ERK1/2 (total and phosphorylated) in TYK-nu (H) and OVCAR4 (I) treated with 0–30 μM carboplatin without or with 50 μM LJH685 for 72 h.

Data information: In (B and F), data are presented as mean (SD). *P < 0.05, **P < 0.01. Exact P-values are provided in Appendix Table S10, Student's t-test.

Source data are available online for this figure.

## RSK activity mediates oncogenic EphA2-S897 phosphorylation in OC cells

To uncover the mechanism of the EphA2-S897 phosphorylation, we used inhibitors against the kinases reported to phosphorylate the receptor in other cancers (Miao *et al*, 2009; Zhou *et al*, 2015; Barquilla *et al*, 2016). As above, cisplatin increased EphA2-pS897 in TYK-nu (Fig 4E). Inhibitors of MEK, upstream of ERK1/2-RSK (UO126), and to less extent PKA (H89.2HCl) reduced the

constitutive and platinum-induced EphA2-pS897, whereas PI3K inhibition, upstream of Akt (LY294002 and αPI3K BYL-719), had minor effects on this phosphorylation (Fig 4E). Most effectively, the RSK inhibitor (RSKi) BI-D1870 blocked EphA2-pS897 in TYK-nu (Fig 4E and F; 58.8 ± 21.2% reduction in the absence of cisplatin, P = 0.003) and TYK-nu.R (Fig 4G).

Moreover, carboplatin triggered the activation of ERK1/2 and RSK, indicated by increased ERK1/2-T202/Y204 and RSK-T359/S363 phosphorylation, in TYK-nu and OVCAR4 (Fig 4H and I). The RSKi

LJH685, with limited off-target effects (Aronchik *et al*, 2014), prevented both the platinum-induced RSK activation and EphA2-pS897 (Fig 4H and I; see Appendix Fig S3A for BI-D1870 and LJH685 comparison). In contrast, upstream ERK1/2-pT202/Y204 was even increased by RSKi (Fig 4H and I). These results identify ERK1/2-RSK axis as the platinum-activated pathway essential for EphA2-S897 phosphorylation in OC cells.

## Inhibition of RSK and EphA2-pS897 enhances EphA2-pY588, concurrently reducing OC cell viability

To test whether the RSK-EphA2 axis inhibition affects platinum resistance, OC cells were treated with cisplatin with and without RSKi. While BI-D1870 alone had minor effects on OVCAR4 and OVCAR8 viability, this treatment significantly sensitized the cells to platinum (Fig 5A; reduction in viability: OVCAR4 52.9 ± 8.2% at 5 μM cisplatin, OVCAR8 78.9 ± 12.7% at 20 μM cisplatin, *P* < 0.001). Consistently, OVCAR8 was sensitized to cisplatin by

LJH685 (Fig 5B, Appendix Fig S3B; 52.3 ± 9.8% reduced viability at 20 μM cisplatin, *P* = 0.011). Platinum-sensitive TYK-nu was less affected by RSKi (Fig 5A and B), whereas the viability of TYK-nu.R was markedly decreased relative to untreated control (Fig 5A; 64.0 ± 11.1%, *P* = 0.002). Further, the combination of platinum with RSKi effectively eliminated TYK-nu.R cells (Fig 5A; reduction in viability: 62.8 ± 33.3% at 10 μM cisplatin, *P* = 0.004).

Despite this close correlation between RSK-EphA2 axis and platinum resistance, and contrary to results with other cell lines (Landen *et al*, 2005a; Shen *et al*, 2013), siRNA-mediated EphA2 knockdown did not significantly alter TYK-nu and TYK-nu.R viability or cisplatin sensitivity in 2D or 3D (Appendix Fig S3C and D). EphA2 depletion increased, however, the poorly characterized EphA2 interacting orphan receptor GPRC5A (Appendix Fig S3C). To understand these results, we further assessed the effects of RSKi in EphA2 in the cells with differential treatment responses. Notably, LJH685 alone and combined with cisplatin increased tumor-suppressive EphA2-pY588 in OVCAR4 and OVCAR8, i.e., the cells

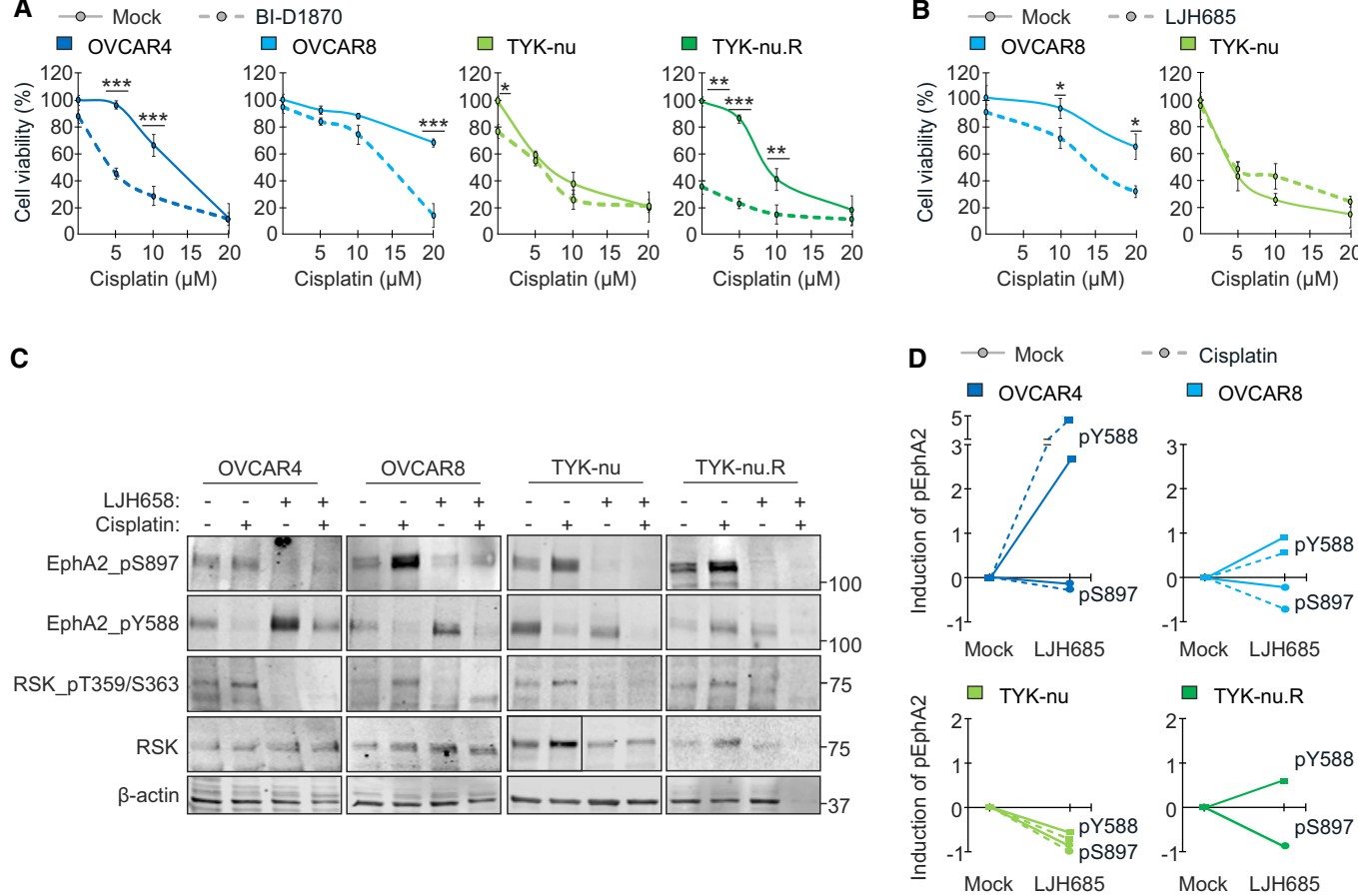

**Figure 5. RSK-EphA2-pS897 inhibition enhances EphA2-pY588 and sensitizes OC cells to platinum chemotherapy.**

A, B Charts show OC cell viability after 72-h treatment with 0–20 μM cisplatin and 10 μM BI-D1870 (A; *N* = 4) or with 50 μM LJH685 (B; *N* = 3) as indicated.

C RSK and EphA2 (total and phosphorylated) in OC cells treated with 10 μM cisplatin and 25 μM LJH685 as indicated for 72 h.

D Charts illustrate the corresponding quantified inhibition of EphA2-pS897 and increase in EphA2-pY588 shown in (C).

Data information: In (A, B), data are presented as mean (SD). *P* < 0.05, **P* < 0.01, ***P* < 0.001. Exact *P*-values are provided in Appendix Table S10, Student's *t*-test.
Source data are available online for this figure.

effectively sensitized to platinum (Fig 5C and D). Consistently, this tumor-suppressive EphA2 axis was most prevalent in the untreated cisplatin-sensitive TYK-nu, while the acute TYK-nu.R sensitivity to LJH685 was coupled with increased EphA2-pY588 (Fig 5C and D; NA after collapsed cell viability by combinatorial treatment).

Considering the EphA2-pS897 inhibition results (see Fig 4E), we also tested the effect of Trametinib, an FDA approved MEKi (Wright & McCormack, 2013; Odogwu *et al*, 2018), in cell responses to platinum. Coincident with ERK1/2-pT202/Y204 inhibition, Trametinib decreased viable OVCAR4 by over 40% compared to untreated control, whereas the relative cell cisplatin sensitivity remained unaltered (Appendix Fig S3E and F). *In vivo,* Trametinib did not affect carboplatin-induced apoptosis or decrease proliferation significantly in our xenograft pilot experiment (Appendix Fig S3G–J). Therefore, rather than broad MEK-ERK1/2 pathway inhibition or EphA2 knockdown, the specific RSK-EphA2-pS897 blockade and consequent reversal to tumor-suppressive EphA2-pY588 correlated with the effective OC cell elimination and sensitization to platinum.

### RSK regulates EphA2-associated orphan receptor GPRC5A, controlling platinum resistance

To clarify the molecular mechanisms governing the EphA2 signaling duality and treatment resistance, we analyzed the poorly characterized EphA2 interactor GPRC5A (Bulanova *et al*, 2017). In OC cells, GPRC5A was detected as 41–46 kDa protein (Fig 6A). Unexpectedly, the 41 kDa form diminished and the 46 kDa form became prominent after treatment with cisplatin or LJH685 in both platinum-resistant, RSKi-sensitive OVCAR8 and TYK-nu.R (Fig 6A and B; $P \leq 0.025$, TYK-nu.R NA after combinatorial treatment due to collapsed cell viability). By immunofluorescence, cisplatin treatment led to partial translocation of the otherwise intracellularly accumulated GPRC5A to cell surface, thus enhancing co-localization with EphA2 (S897-phosphorylated; see Fig 5C and D) in TYK-nu.R (Fig 6C, Appendix Fig S4). Instead, RSK-EphA2-pS897 inhibition by BI-D1870 led to EphA2-pY588 induction (see Fig 5C and D) coupled with EphA2 downregulation with and without cisplatin, thus impairing the cisplatin-induced receptor co-localization (Fig 6C, Appendix Fig S4).

In TYK-nu, platinum-mediated increase in cell surface EphA2-pS897 instead correlated with intracellular GPRC5A localization (Fig 6C, Appendix Fig S4; see Fig 5C and D for EphA2-pS897 increase). Moreover, siRNA-mediated EphA2 depletion resulted in stronger GPRC5A upregulation in TYK-nu than in TYK-nu.R (Appendix Fig S3C). Taken together with EphA2 upregulation (total and pS897) after GPRC5A partial knockdown upon platinum

treatment in TYK-nu, but not in TYK-nu.R (Appendix Fig S5A–C), these results suggest that EphA2 and GPRC5A are mutually negatively regulated in the platinum-sensitive cells. In cisplatin-resistant TYK-nu.R, the platinum-induced GPRC5A-EphA2-pS897 co-regulation was in turn impaired by RSKi through EphA2-pY588 activation coupled to EphA2 downregulation, coincident with the effective treatment response.

Consistent with the RSKi results, knockdown of RSK1 and/or RSK2, which are the cancer-associated RSKs highly expressed in OC cells (Fig EV3A), inhibited the platinum-induced EphA2-pS897 (Fig 6D and E). Coincidentally, the viability of cisplatin-treated TYK-nu.R was reduced (Fig 6F; control siNT $24.9 \pm 11.2\%$ vs. siRSK1 $9.2 \pm 4.2\%$, siRSK2 $8.3 \pm 6.6\%$, siRSK1/2 $8.0 \pm 8.0\%$, $P \leq 0.016$). In OVCAR4 and OVCAR8, the EphA2 oncogenic switch was less affected by RSK1 knockdown, but effectively reversed to EphA2-pY588 after RSK2 knockdown (Fig EV3B). Notably, RSK1 depletion effectively suppressed GPRC5A (46 kDa) upon platinum treatment in the resistant OVCAR8 and TYK-nu.R, coincidentally enhancing apoptosis, indicated by increased cleaved PARP in TYK-nu.R (Figs 6E and EV3B). In the absence of cisplatin, RSK2 depletion increased cleaved PARP, whereas the proliferation marker PCNA was generally less affected by RSK1/2 knockdown (Fig 6E).

Moreover, RSK1 or GPRC5A overexpression increased OVCAR4 and OVCAR8 viability after cisplatin treatment (Fig EV3C–E). Overexpression of EphA2 and RSK2 to some extent also enhanced OVCAR4 resistance to cisplatin (Fig EV3C and F; $P \leq 0.048$ at 5 μM cisplatin). Therefore, cisplatin activated the EphA2 tyrosine–serine phosphorylation switch through RSK(1/)2 coincident with RSK1 (/2)-dependent GPRC5A co-regulation to promote OC cell evasion from platinum-induced apoptosis.

### Combinatorial RSKi-platinum treatment induces apoptosis in 3D co-culture and *in vivo*

EphA2 signaling alters cell interactions with the TME (Gucciardo *et al*, 2014; Zhou & Sakurai, 2017). Therefore, to assess the regulation of apoptosis in the context of relevant cell and ECM interactions, we generated mono- and co-culture spheroids of OVCAR8-RFP and patient-derived cancer-associated fibroblasts (OCKI_p22 CAF) and cultured them in 3D collagen. In OVCAR8-RFP monocultures, cisplatin treatment for 20 h enhanced apoptosis (clCasp3), whereas clCasp3 remained less affected by LJH685 alone (Fig 7A and B, and Appendix Fig S6A). Of note, apoptosis was further increased after combining LJH685 with cisplatin (Fig 7A and B). In CAF monocultures, apoptosis remained unaltered by the treatments (Fig 7A and B). Notably, in 3D co-culture with CAFs, cisplatin as

---

**Figure 6. GPRC5A regulation defines OC cell responses to RSK1/2-EphA2-pS897 inhibition.**

A, B  Immunoblot images (A) and quantification (B) show GPRC5A 46/41 kDa ratios in OC cells treated with 0–10 μM cisplatin and with 25 μM LJH685 alone or as a combination for 72 h. The ratio in mock cells was set to 1. *N* = 4.

C  Confocal micrographs of EphA2 (red) and GPRC5A (green) in TYK-nu and TYK-nu.R treated with 5 μM cisplatin and 10 μM LJH685 as indicated for 72 h. Arrows point co-localization of the receptors. Scale bars: 20 μm.

D, E  EphA2 (total and phosphorylated), RSK1/2, and GPRC5A in TYK-nu (D) and these proteins coupled with PARP and PCNA in TYK-nu.R (E) were assessed by immunoblotting after RSK1/2 knockdown for 2 days and following treatment with 0–5 μM cisplatin for 72 h. Asterisks indicate unspecific bands.

F  Chart illustrates the viability of the RSK1/2-depleted TYK-nu.R after cisplatin treatment. *N* = 3.

Data information: In (B and F), data are presented as mean (SD). \*$P < 0.05$, \*\*$P < 0.01$. Exact *P*-values are provided in Appendix Table S10, Student's *t*-test.
Source data are available online for this figure.

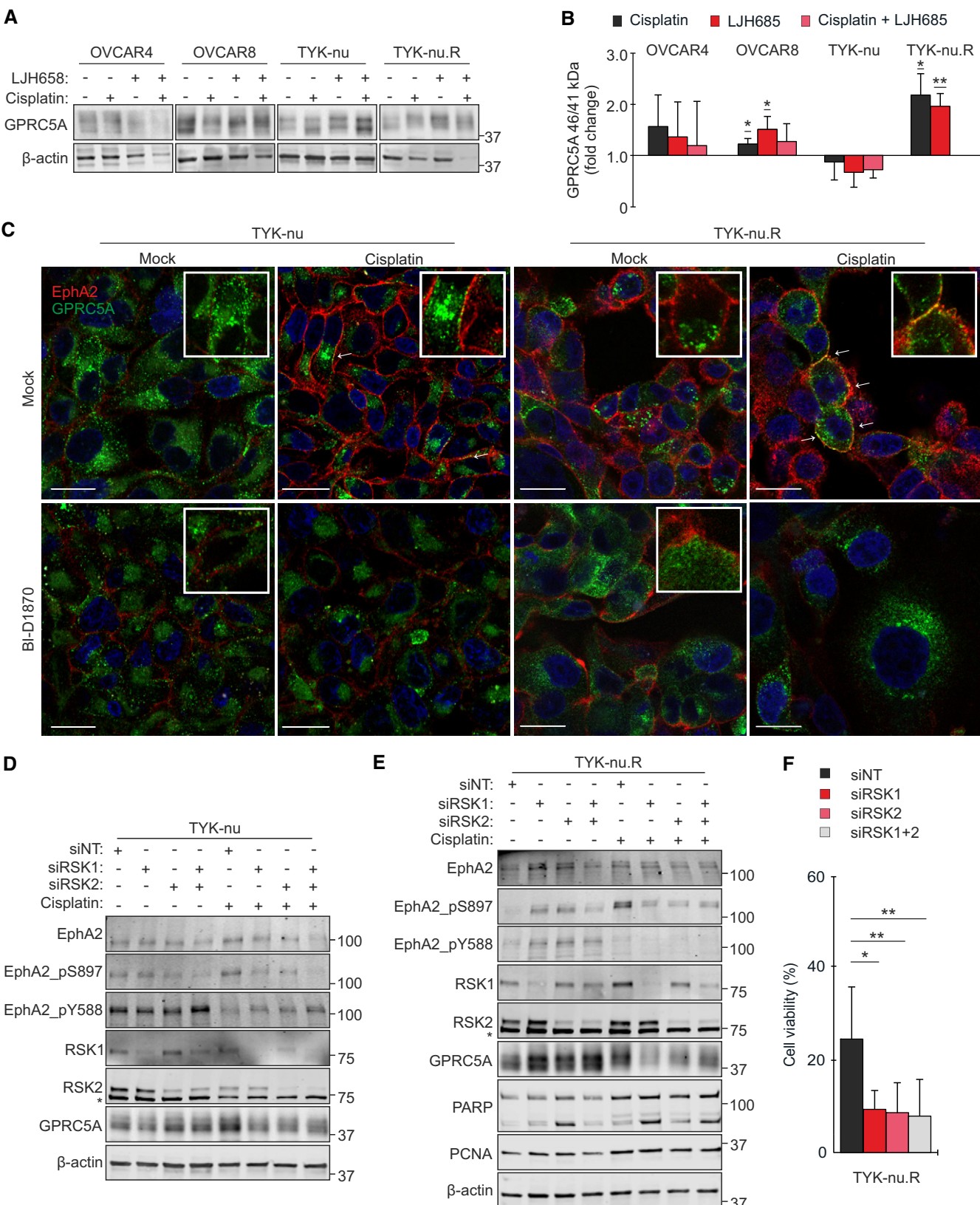

**Figure 6.**

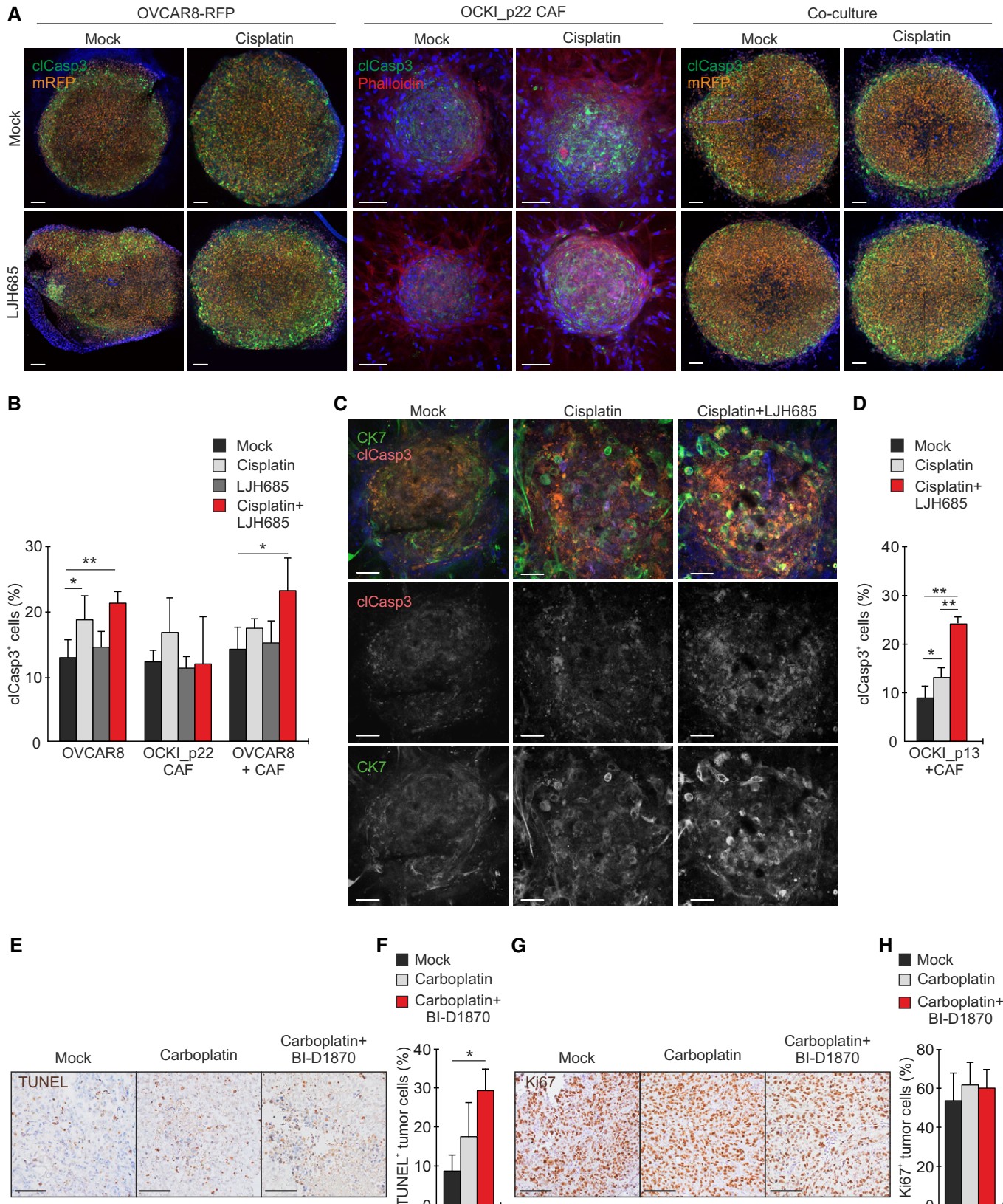

**Figure 7.**

**Figure 7.  RSKi combinatorial treatment induces apoptosis in 3D co-culture and *in vivo*.**

A       Confocal micrographs show cleaved caspase-3 (clCasp3, green), mRFP (orange), and phalloidin (F-actin, red; only shown for CAF) in 3D OVCAR8-RFP and OCKI_p22
        CAF mono- and co-cultures treated without or with 20 μM cisplatin or 25 μM LJH685 alone or in combination for 20 h. Scale bars: 100 μm.
B       Quantitative assessment of clCasp3⁺ cells in the 3D cultures. N = 4.
C       Confocal micrographs show cytokeratin 7 (CK7, green) and phalloidin (F-actin, red) in 3D patient OCKI_p13 mono- and co-culture with OCKI_p22 CAF treated
        without or with 20 μM cisplatin or 25 μM LJH685 alone or in combination for 20 h. Scale bars: 50 μm.
D       Quantitative assessment of clCasp3⁺ OCKI_p13 cells in 3D co-cultures of patient-derived cells. N = 4.
E–H     Immunohistochemistry images and quantifications of TUNEL (E, F) and Ki67 (G, H) in OVCAR4 xenografts in mock (N = 4)-, carboplatin (N = 4)-, and
        carboplatin + BI-D1870 (N = 5)-treated mice. Scale bars: 100 μm.

Data information: In (B and D, F, and H), data are presented as mean (SD). *P < 0.05, **P < 0.01. Exact P-values are provided in Appendix Table S10, Student's t-test
(B, D) and Mann–Whitney U-test (F, H).

well as LJH685 alone failed to induce OVCAR8-RFP apoptosis, whereas the combinatorial LJH685-cisplatin treatment effectively increased clCasp3 (Fig 7A and B; $P = 0.022$).

Importantly, combination of LJH685 with cisplatin further enhanced platinum-induced apoptosis in the spheroid co-cultures of patient-derived OCKI_p13 HGSC cells and OCKI_p22 CAFs in 3D collagen (Fig 7C and D; see Appendix Fig S6B for morphological characterization of the patient HGSC 3D mono- and co-cultures). *In vivo*, combinatorial RSKi-platinum treatment of the OVCAR4 xenografts for 48 h likewise increased apoptosis significantly (Fig 7E and F; tumor TUNEL increased $2.5 \pm 1.8$-fold, $P = 0.014$), whereas proliferation (Ki67) remained unaltered (Fig 7G and H).

## Patient-derived GPRC5A^high HGSC cells are sensitive to combinatorial RSKi-platinum treatment

To validate the clinical relevance of our findings, we next assessed RSK, EphA2, and GPRC5A by immunofluorescence in human HGSC tumor tissue sections. Notably, all these proteins showed higher expression in HGSC than in the stroma (Fig 8A, Appendix Fig S7A). By immunoblotting of patient-derived cells, EphA2 (total and phosphorylated), GPRC5A, and RSK showed variable expression in the cancer cells, whereas GPRC5A was undetectable and RSK low in patient-derived CAFs (Fig 8B). As expected, CAFs expressed PDGFRβ and FSP1, and cancer cells E-cadherin and PAX8, whereas both cell types were positive for vimentin and N-cadherin (Fig 8B).

To assess the specificity of the RSKi-platinum response, HGSC patient-derived cancer and stromal cells as well as normal fibroblasts were treated with RSKi alone and combined with cisplatin. Notably, cisplatin increased EphA2 and pS897/pY588, and BI-D1870 blocked both the constitutive and cisplatin-induced EphA2-pS897 in the HGSC cells with high GPRC5A, concurrently increasing tumor-suppressive EphA2-pY588 (Fig 8C and D, Appendix Fig S7B). Treatment with LJH685 likewise reverted the platinum-induced EphA2 switch (Appendix Fig S7C). Coincident with this signaling reversal, RSKi significantly sensitized the cells to platinum (Fig 8E; reduction in viability at 20 μM cisplatin: OCKI_p02 $68.1 \pm 40.1\%$; OCKI_p06 $48.7 \pm 21.4\%$, $P \leq 0.027$). In contrast, RSKi neither reverted efficiently the platinum-induced EphA2 switch in cells with low GPRC5A (Appendix Fig S7D and F) nor altered the viability of patient-derived CAFs, mesothelial cells, or CCL-137 embryonic lung fibroblasts (Fig 8F and G, Appendix Fig S7I). While cisplatin treatment variably induced and RSKi reduced EphA2-pS897, EphA2-pY588 was not restored upon cisplatin treatment in the benign cells (Fig 8H, Appendix Fig S7G, H and J).

Notably, GPRC5A localization in *ex vivo* 3D cultures of the RSKi-sensitive HGSC cells resembled the corresponding pattern in TYK-nu.R, including the platinum-induced translocation to cell surface, leading to co-localization with EphA2 (Fig 8I; see Fig 6C for TYK-nu.R). The EphA2-GPRC5A co-localization was likewise increased in the RSKi-sensitive xenograft tumors after carboplatin treatment *in vivo* (Fig EV4A and B; $7.2 \pm 0.7$-fold increase, $P = 0.0002$). Altogether, these results reinforce the link between cancer cell-expressed

**Figure 8.  RSK-EphA2-pS897 inhibition specifically sensitizes GPRC5A^high HGSC cells to platinum chemotherapy.**

A       Representative confocal micrographs show EphA2 (red), RSK (green: top), and GPRC5A (green: bottom) in frozen sections of HGSC patient tumors. S indicates the
        stroma. Scale bars: 50 μm.
B       Indicated proteins and EphA2 phosphorylation were assessed by immunoblotting in patient-derived HGSC cells and CAFs.
C       GPRC5A in earlier (top) and later passage (bottom) HGSC cells. See normalized GPRC5A relative to OCKI_p01 below the immunoblot images.
D       Charts illustrate EphA2-pS897 inhibition (circles) and EphA2-pY588 increase (squares) by 10 μM BI-D1870 alone (unbroken line) or in combination with 0–10 μM
        cisplatin (dotted line) in the GPRC5A^high HGSC cells. See Appendix Fig S7B for immunoblots.
E       Cell viability of the GPRC5A^high OCKI_p02 and OCKI_p06 after treatment with a combination of 0–20 μM cisplatin and 10 μM BI-D1870 for 72 h. N = 3.
F, G    Cell viability upon 72-h combinatorial treatment with 0–20 μM cisplatin and 25 μM LJH685 in patient-derived CAFs (F; N = 3) and 10 μM BI-D1870 in mesothelial
        cells (G; N = 3).
H       Chart illustrates the changes in EphA2-pS897 and EphA2-pY588 by BI-D1870 alone and with cisplatin in mesothelial cells. Upon combination with 10 μM cisplatin
        for 72 h, 10 μM BI-D1870 decreased EphA2-pS897 and EphA2-pY588. See Appendix Fig S7J for immunoblot.
I       Confocal micrographs of EphA2 (red) and GPRC5A (green) in primary OCKI_p06 patient cells cultured in 3D collagen for 4 days before 72-h treatment with or
        without 10 μM cisplatin. Scale bars: 20 μm.

Data information: In (E–G), data are presented as mean (SD). *P < 0.05. Exact P-values are provided in Appendix Table S10, Student's t-test.
Source data are available online for this figure.

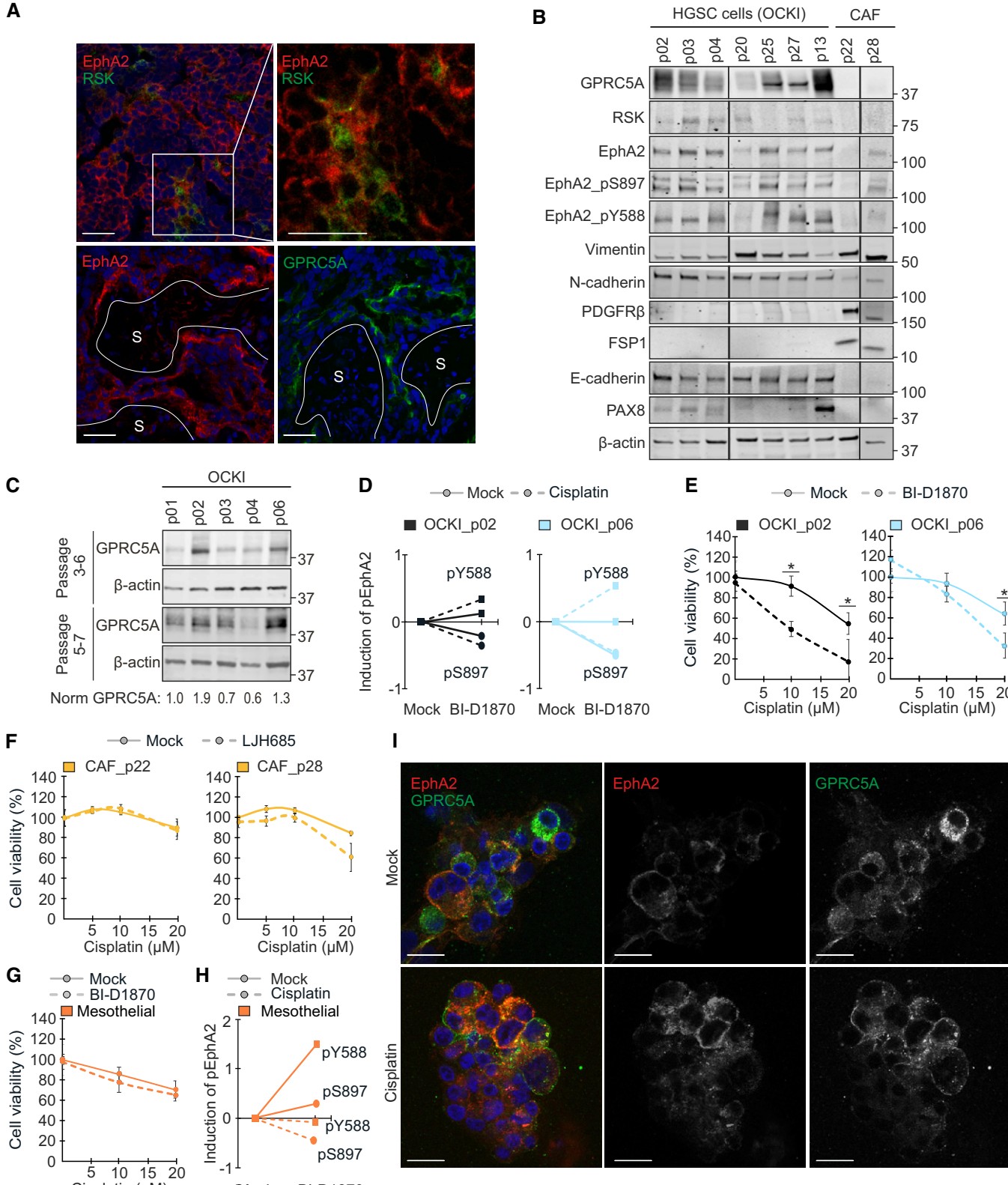

**Figure 8.**

GPRC5A, the RSK1/2-EphA2 signaling inhibition, and treatment sensitivity of HGSC cells.

### High GPRC5A is associated with poor survival and chemotherapy resistance in HGSC patients

Since GPRC5A remains uncharacterized in clinical HGSC, we performed histological analysis of tissue microarrays (TMA) containing samples from primary and metastatic tumors of 126 treatment-naïve HGSC patients (Fig 9A, Appendix Table S1). The maximum GPRC5A intensity in cancer cells was classified as low (negative, mild, or moderate) or high (Fig 9B). There was no significant correlation between GPRC5A intensity in primary vs. metastatic cores, indicating that GPRC5A protein expression was independent of the tumor site (Appendix Table S2). The GPRC5A grading did not correlate with clinico-pathological variables except for FIGO stage in primary tumors (Appendix Table S3; $P = 0.047$). Thus, the metric maximum intensity of GPRC5A in cancer cells was not dependent on most analyzed variables.

To examine the correlation between GPRC5A and patient outcome, we performed survival analyses for primary and metastatic tumors separately. In both groups, high GPRC5A was significantly associated with worse overall survival (OS) and with shorter progression-free survival (PFS) in the metastatic tumors (Fig 9C and D, Appendix Table S4; OS: $P = 0.044$ and $0.012$, PFS: $P = 0.009$). By univariate analysis, all clinico-pathological variables (except for FIGO stage and type of surgery) and GPRC5A were significantly associated with worse OS in both primary and metastatic cases (Appendix Table S5). Additionally, GPRC5A was associated with shorter PFS in the metastatic tumors (Appendix Table S6). In line with these survival results, Cox multivariate analysis validated the association of GPRC5A with worse OS in both groups (Appendix Table S5), whereas shorter PFS was significantly associated with GPRC5A only in metastatic cases (Appendix Table S6).

Of note, GPRC5A in primary tumors was inversely correlated with the reduction in tumor burden at the end of treatments, defined by the objective response rate (ORR), and with platinum sensitivity, considering sole administration or combination with taxane (Fig 9E, Appendix Table S7; $P = 0.034$ and $0.011$). In The Cancer Genome Atlas (TCGA) OC mRNA dataset, GPRC5A was associated with worse OS survival (Fig EV5A; 40% highest vs. 40% lowest GPRC5A expressing patients, $P = 0.045$) and co-expressed with EphA2, for which we did not obtain reliable detection in the TMA (Fig EV5B; http://cancergenome.nih.gov/). Using an independent publicly available cohort with mRNA data from 125 treatment-naïve HGSC patients (Pils et al, 2012), we further validated the potential of combined GPRC5A and EphA2 expression to predict patient PFS (Fig 9F–H; $P = 0.020$, NS for EphA2 or GPRC5A alone). The mRNA for RSK1-4 did not improve the significance of this predictive power (Figs 9I and EV5C–F). These results indicate that the GPRC5A-EphA2 axis can predict poor platinum-based chemotherapy responses and shorter survival in patients with HGSC.

## Discussion

Surgery and platinum-based chemotherapy, the standard of care for HGSC patients, often efficiently eliminate macro-metastases and cancer cells accumulated in ascites. Yet micro-metastases remain, allowing recurrence of increasingly resistant disease, which is a major challenge for successful clinical care. Here, using established chemo-sensitive and chemo-resistant HGSC cell models in vitro and in vivo, as well as treatment-naïve primary and early passage ex vivo cultures from HGSC patients, we uncovered a robust mechanism, whereby cancer cells gain platinum resistance via the treatment-induced, adaptive RSK1/2-EphA2-GPRC5A signaling switch.

Clinical evidence indicates that even after initial drug response, majority of relapsed HGSCs repeatedly respond to platinum-based chemotherapy (Pfisterer & Ledermann, 2006). Therefore, improved understanding of the signaling pathways governing the phenotypic plasticity of the resistant micro-metastatic cells, rather than focus only on the emergent genetically chemo-resistant clones, can provide new strategies to develop more effective treatments. Indeed, signals for cancer invasiveness and stemness as well as epithelial-to-mesenchymal transition have been suggested as mechanisms for increased chemoresistance (Diepenbruck & Christofori, 2016), yet how such pathways operate and evolve in the treatment-resisting cells upon chemotherapy remains unclear. Our results identify one such mechanism via ERK1/2-RSK1/2 kinase pathway activated in OC cells by platinum chemotherapy. Besides DNA damage, platinum is known to induce oxidative stress/ROS-related ERK1/2 activation in different types of malignant and non-malignant cells (Dasari & Tchounwou, 2014). Based on our results, this will lead to EphA2-S897 phosphorylation, previously shown to mediate stemness, invasion, and metastasis in different types of cancer (Zhou & Sakurai, 2017).

After encouraging pre-clinical data showing reduced tumor growth and increased taxane response after EphA2 depletion (Landen et al, 2005b; Shen et al, 2013; Petty et al, 2018), a clinical trial is testing direct EphA2 inhibition in advanced metastatic cancers (NCT01591356, https://clinicaltrials.gov). We found, however, that the effective RSKi treatments were tightly coupled with the induction of tumor-suppressive EphA2-pY588. Therefore, our results suggest that instead of ablation of both tumor-suppressive and oncogenic receptor activities, the reversal of the EphA2 oncogenic signaling switch may better correlate with effective HGSC sensitization to platinum-based chemotherapy.

In our HGSC cell cultures and ex vivo models, pharmacological RSK inhibition blocked the activation of the tumor-promoting EphA2 activity, which in turn led to enhanced canonical tumor-suppressive EphA2-pY588 signaling, coincidentally sensitizing the cells to platinum-induced apoptosis. These results help to explain the previous findings that RSK2 gene silencing increases OC cell sensitivity to cisplatin, and RSK inhibition with BI-D1870 reduces tumor growth and improves survival in xenograft models (van Jaarsveld et al, 2013; Hammoud et al, 2016). Furthermore, our knockdown results reveal specific contributions of RSK1 and RSK2 in the oncogenic switch, including active RSK1 function in the platinum-induced GPRC5A regulation coupled to apoptosis evasion of the platinum-resistant cancer cells. This is noteworthy since CAFs lacking GPRC5A remained unaffected by RSKi addition to cisplatin treatment, whereas the combinatorial treatment specifically induced OC cell apoptosis in 2D and 3D. While no RSKi has entered clinical trials, possibly due to off-target effects and toxicity or suboptimal pharmacokinetic properties, the upstream MEKi Trametinib is in clinical use for certain other cancers (Faghfuri et al, 2018). In our cell and xenograft pilot experiments, Trametinib–carboplatin combination did

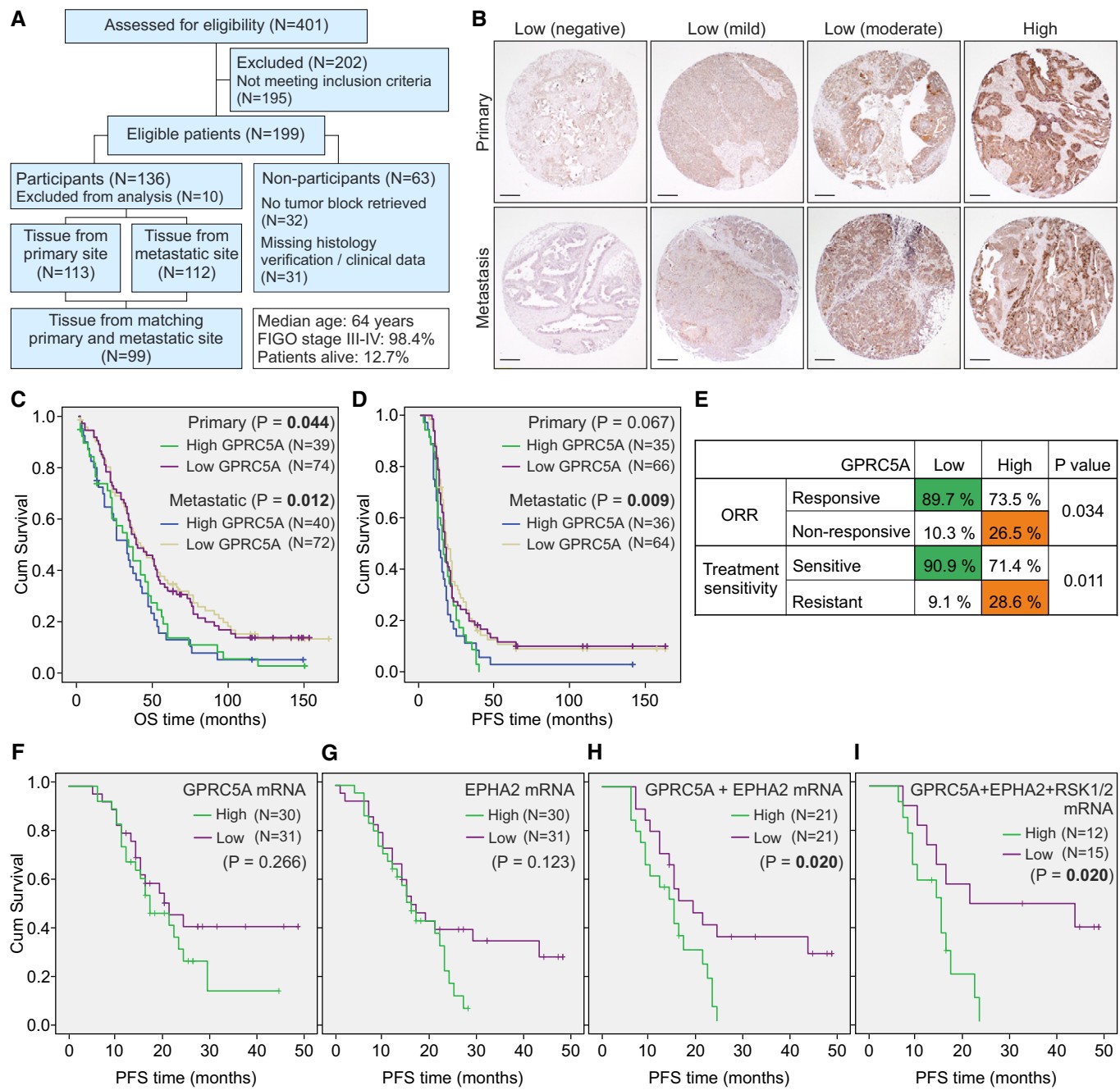

**Figure 9. High GPRC5A is associated with poor survival and chemotherapy resistance in HGSC patients.**

A    Flow diagram of patients for the HGSC TMA.

B    Representative immunostainings for GPRC5A scoring in HGSC primary and metastatic tumors. For maximum intensity scores, an optical 4-point scale (0: negative, 1: mild, 2: moderate, and 3: high) was used. For survival analyses, a 2-point scale (low: scores 0–2 and high: score 3) was used. Scale bars: 200 µm.

C, D    Kaplan–Meier survival curves illustrate the overall (OS) and progression-free (PFS) survival of patients with high or low GPRC5A. In primary tumors, mean OS for GPRC5A$^{high}$ was 39 months vs. 56 months for GPRC5A$^{low}$ (C). In patients with metastatic tumors, mean OS for GPRC5A$^{high}$ was 37 months vs. 60 months for GPRC5A$^{low}$. Mean PFS in metastatic tumors was 20 months for GPRC5A$^{high}$ vs. 35 months for GPRC5A$^{low}$ (D). No significant difference in mean PFS in primary tumors.

E    Correlation of maximum GPRC5A intensity with objective response rate (ORR) and treatment sensitivity (platinum alone or in combination with taxane).

F–I    Kaplan–Meier survival estimates for the PFS of patients with high or low (top 25% vs. bottom 25%) GPRC5A (F), EPHA2 (G), GPRC5A + EPHA2 (H), and GPRC5A + EPHA2 + RSK1(RPS6KA1) + RSK2(RPS6KA3)-combined (I) mRNA expression. Mean PFS for GPRC5A + EPHA2$^{high}$ was 15 months vs. 26 months for GPRC5A + EPHA2$^{low}$ (H). Mean PFS for GPRC5A + EPHA2 + RPS6KA1/3$^{high}$ was 14 months vs. 31 months for GPRC5A + EPHA2 + RPS6KA1/3$^{low}$ (I).

Data information: In (C, D and F–I), logrank test was used. In (E), Pearson chi-square test was used.

not, however, induce more apoptosis than carboplatin alone, suggesting different modes of MEKi action from the specific EphA2-GPRC5A signaling reversal by RSKi-platinum combination. Since MEKi can trigger broad effects in, e.g., cell cycle regulation, possibly hindering the RSK-dependent cancer cell sensitization to platinum cytotoxicity, our results warrant further RSKi investigation and development for a platinum combination.

Another challenge for improved HGSC treatments is the lack of biomarkers for disease aggressiveness and specific treatment responses. Currently, BRCA1/2 mutations, which deprive the cells from efficient homologous recombination (HR) DNA repair, to some extent predict platinum chemotherapy response and are used to stratify patients for promising combinatorial PARP inhibitor treatments (Ledermann et al, 2016; Coleman et al, 2017). Moreover, functional HR deficiency assays have been able to score HR deficiency ex vivo, predicting not only platinum response but also patient survival (Tumiati et al, 2018). We found that the platinum-induced oncogenic RSK1/2-EphA2 signaling switch was coupled to EphA2-GPRC5A co-localization. As yet poorly understood, the context-dependent functions of GPRC5A include facilitating HR DNA repair, cell proliferation, and migration, as well as cell survival under hypoxia (Sokolenko et al, 2014; Greenhough et al, 2018; Liu et al, 2018). Such functions could be regulated by GPRC5A phosphorylation, since crosstalk of this receptor with EGFR induces GPRC5A phosphorylation and switch from tumor-suppressive to oncogenic activities in lung cancer (Lin et al, 2014; Wang et al, 2016). In our experiments, RSK1 depletion impaired the platinum-dependent GPRC5A regulation coincident with increased apoptosis (cleaved PARP). Since GPRC5A can be transcriptionally suppressed by TP53, which is mutated in essentially all HGSCs, GPRC5A induction can occur frequently in OC (Wu et al, 2005; Domcke et al, 2013). Our results from histological analysis of GPRC5A in primary and metastatic tumor specimens of advanced HGSC cohort, coupled to data from publicly available mRNA expression datasets, revealed the potential of this receptor as a biomarker for HGSC abysmal outcome and poor chemotherapy response, and also for aiding at the stratification of the unresponsive patients for treatment sensitization through novel combinatorial therapies, such as the RSK1/2-EphA2-pS897 pathway inhibition.

To conclude, we have discovered a robust platinum-induced oncogenic signaling switch along RSK1/2-EphA2-GPRC5A axis which contributes to chemotherapy-driven adaptive resistance in HGSC. Our results raise the question whether combinatorial targeted approaches such as RSK inhibition would enable overcoming OC adaptive platinum resistance. Additionally, our findings on GPRC5A predicting poor survival and treatment response could potentially fill in for the gap of clinically applicable biomarkers that are greatly needed to improve OC treatment scheme.

# Materials and Methods

## Study approval

The Swedish Ethical Review Agency (Etikprövningsmyndigheten) approved the collection of samples from patients with metastatic HGSC (2016/1197-31/1, 2016/2060-32) and the collection of tumor tissue for the TMA study (2012/539-31/1). In both cases, written informed consent was received from participants prior to inclusion in the studies. All experiments were performed according to the principles set out in the World Medical Association Declaration of Helsinki and the Department of Health and Human Services Belmont Report. OC tumor xenograft experiments were approved by the National Animal Experiment Board in Finland (ESAVI/8983/04.10.07/2015) and the Animal Experiment Board in Osaka City University in Japan (19001) and performed in compliance with ethical regulations for animal experiments and welfare.

## Patient samples

Abdominal ascites fluid and omental tumors were collected at the Karolinska University Hospital (see Table 1). Isolation of HGSC cells from ascites fluid was performed immediately after acquisition. After centrifugation at 3,200 g for 10 min at 4°C, the supernatant was filtered using a 0.22-μm strainer to obtain cleared fluid used as supplement for cell culture media. If needed, red blood cells were lysed from the cell pellet by using Tris-buffered ammonium chloride solution (Tris-NH4Cl). Multicellular clusters were collected from the cell suspension using 45-μm strainer and re-suspended in 1:1 DMEM:F12 medium with 100 mg/ml penicillin/streptomycin and 10% clarified ascites. Cells were cultured at 37°C in 5% $CO_2$ incubator and routinely checked using MycoAlertPlus™ Mycoplasma Detection Kit (Lonza). All experiments were performed in complete media (i.e., containing clarified ascites). For CAF isolation, fresh omental tumors were dissected into small explants and cultured on collagen type I (50 μg/ml, Sigma)-coated dishes in fibroblast medium (Fibroblast Medium Kit, Innoprot). The patient-derived cells were characterized by immunofluorescence of epithelial and stromal markers immediately after isolation and during culture.

The characterization of cell cultures was corroborated by p53 immunofluorescence of mock- and nutlin-treated cells (16-h treatment with 10 μM nutlin), considering that nutlin selectively stabilizes wild-type TP53 and that majority of HGSC have TP53 mutations (Domcke et al, 2013). Patient-derived CAFs and ARN8 wild-type TP53 cells as well as mutant TP53 OVCAR8, TYK-nu, and TYK-nu.R were used as controls. HGSC cells that responded to nutlin treatment (wild-type TP53) were classified as mesothelial cells and cultured as above.

## Cell lines

NIH:OVCAR3, OVCAR4, and OVCAR8 (National Cancer Institute, USA) were maintained in RPMI and TYK-nu and TYK-nu.R (Japanese Collection of Research Bioresources Cell Bank; Japan) in MEM (all classified as highly likely or likely human HGSC cell lines; Domcke et al, 2013). ARN8 melanoma cells and CCL-137 fibroblasts (American Type Culture Collection; USA) were maintained in DMEM. All culture media were supplemented with 10% FBS, 100 mg/ml penicillin/streptomycin, and 10 μg/ml insulin (NIH:OVCAR3 only). The cells were cultured and checked for mycoplasma as above.

## Cell spheroids and 3D collagen matrices

For the mono- and co-culture spheroids, OVCAR8 cells were transfected to express recombinant histone H2A-red fluorescent protein

(OVCAR8-RFP). Pure pool of RFP-positive colonies was collected by fluorescent-activated cell sorting and seeded with patient-derived CAFs (1:5 ratio, $3 \times 10^5$ cells/ml) on ultra-low attachment 96-well plates (Corning) and incubated for 48 h at 37°C before embedding in 3D collagen. Mono- and co-culture spheroids of patient-derived HGSC cells with patient-derived CAFs were generated as above.

Rat tail collagen type I was dissolved in 0.3% acetic acid to 4.5 mg/ml stock and diluted to final concentration of 2.25 mg/ml in 2× MEM. pH was adjusted to 7.5–8 with sodium hydroxide. Cells ($1.75 \times 10^6$ cells/ml) and preformed spheroids were embedded in 40 µl collagen (in 10 µl for cytotoxicity assays) and incubated at 37°C up to 7 days before subjecting cells to cytotoxicity assays or immunofluorescence.

### siRNA knockdown and cDNA overexpression

siRNA against human EphA2 (GE Healthcare, Dharmacon: L-003116-00-0005), GPRC5A (QIAGEN: GPRC5A#2 SI00058604 and GPRC5A#5 SI02225734; Appendix Table S9), RSK1 (RPS6KA1; Dharmacon: L-003025-00-0005), RSK2 (RPS6KA3; Dharmacon: L-003026-00-0005), and non-targeting control siRNA (QIAGEN: SI03650318) were transfected in cells using Lipofectamine 2000 (Thermo Scientific).

Cells were transfected with pcDNA3.1-His-V5 vector encoding human EphA2 (Sugiyama et al, 2013), pKH3-human RPS6KA1/RSK1 plasmid (gift from J. Blenis; Addgene plasmid #13841; http://n2t.net/addgene:13841; RRID: Addgene_13841; Richards et al, 2001), pCMV3 N-HA-tagged human RPS6KA3/RSK2 plasmid (Sino Biological), and corresponding empty vectors for mock controls by using FuGENE HD (Promega). For GPRC5A stable overexpression, 293-FT cells were transfected with pLenti-C-Myc-DDK vector containing GPRC5A cDNA, VSV-G envelope plasmid, and pCMV-d8.91 packaging plasmid using Lipofectamine 2000. The medium was changed 6 h after transfection, and the viral supernatants were collected after 4 days, passed through a 0.45-µm filter and used for transduction; controls were generated with the empty vector.

### Drug treatment, inhibitor treatment, and cytotoxicity assay

Inhibitors were purchased from SelleckChem (H89.2HCl, BI-D1870, LJH685, Trametinib) or Cell Signaling Technologies (UO126, LY294002, BYL-719) and diluted in DMSO. Refametinib (Chemietek) was diluted in water.

For combinatorial treatments, cells were treated with the indicated inhibitors for 30 min before addition of cisplatin (Sigma) or carboplatin (SelleckChem). Cytotoxicity was assessed after 72 h using CellTiter-Glo Luminescent Cell Viability Assay (Promega) for 5–20 min before luminescence detection.

### Antibodies

The antibodies used were as follows: primary antibodies against cleaved caspase-3 (Asp175 Alexa Fluor® 488 Conjugate, immunofluorescence (IF) 1:50), E-cadherin (24E10, #3195S, immunoblotting (WB) 1:1,000), EphA2_pS897 (#6347, WB 1:750, IF 1:100), EphA2_pY588 (#12677, WB 1:750), ERK1/2 (#9107S, WB 1:500), ERK1/2_pT202/Y204 (#9101, WB 1:500), PARP (46D11, #9532, WB 1:1,000), PCNA (D3H8P, #13110, WB 1:1,000), RSK1 (D6D5, #8408,

WB 1:1,500), RSK2 (D21B2 XP, #5528, WB 1:1,500), RSK1/2/3 (#9355, WB 1:750, IF 1:25), and RSK_pT359/S363 (#8753, WB 1:750) all from Cell Signaling Technologies. Primary antibodies were purchased from Santa Cruz Biotechnology against EphA2 (C-3, # sc-398832, WB 1:750), p53 (DO-1, # sc-126, IF 1:200), PDGFRβ (D6, # sc-374573, WB 1:500), vimentin (V-9, # sc-6260, WB 1:2000, IF 1:100), β-actin (C-4, # sc-47778, WB 1:2,000), and β-tubulin (D-10, # sc-5274, WB 1:500). Primary antibodies against CK7 (OV-TL 12/30, Invitrogen, MA5-11986, WB 1:1,000, IF 1:100), EphA2 (ECD, R&D Systems, #AF3035, IF 1:100), FSP1 (AT1C3, LifeSpan Biosciences, LS-C755562, WB 1:750), GAPDH (Sigma, Atlas Antibodies, # G8795, WB 1:15,000), GPRC5A (Sigma, Atlas Antibodies HPA007928, WB 1:1,000, IF and immunohistochemistry (IHC) 1:250), Ki67 (Leica, Biosystems, #ACK02, IHC 1:200), N-cadherin (BD Transduction Laboratories, # 610920, WB 1:1,000), and PAX8 (Proteintech, #10336-1-AP, WB 1:2500, IF 1:100) were used.

### Immunoblotting

Cells were lysed on ice with RIPA lysis buffer (50 mM Tris–HCl pH 7.4, 150 mM NaCl, 1% Igepal CA-630, 0.5% sodium deoxycholate) supplemented with 5 mM EDTA, protease inhibitor (cOmplete ULTRA tablet, Sigma), and phosphatase inhibitor (PhosSTOP tablet, Sigma). The lysates were cleared by centrifugation at 21,130 g for 15 min at 4°C, and protein concentrations were determined using the Pierce BCA Assay (Thermo Scientific). Lysates were mixed with 5× sample buffer (0.3 M Tris–HCl pH 6.8, 50% glycerol, 10% sodium dodecyl sulfate, 0.05% bromophenol blue) containing 0.5 M dithiothreitol and heat-denatured at 95°C for 10 min before separation in 4–20% Mini-PROTEAN TGX Precast Protein Gels (Bio-Rad) and transferred to Trans-Blot Turbo Mini Nitrocellulose Transfer Packs (Bio-Rad). Membranes were blocked for 45 min with 5% milk (Cell Signaling Technologies) or 3% fish gelatin (Sigma) in Tris-buffered saline (TBS; 10 mM Tris, pH 7.6, 150 mM NaCl) and probed with primary antibody in TBS 0.1% Tween-20 (TBS-T) with 5% milk or 3% fish gelatin at the recommended dilutions at 4°C overnight. Membranes were incubated with horseradish peroxidase (HRP)-conjugated secondary antibodies (Dako) or with IRDye Subclass-Specific Antibodies (LI-COR Biosciences) diluted in TBS-T for 1 h at RT, and the signal was detected using ECL chemiluminescent detection reagent (GE Healthcare) and visualized using ChemiDoc Imaging System (Bio-Rad) or using Odyssey Imaging System (LI-COR Biosciences).

### Metastatic *in vivo* ovarian cancer model

293-GPG cells were transfected with the pMx-*Renilla* luciferase–GFP fusion reporter plasmid using Lipofectamine 2000. Viral supernatants were obtained and processed as described above and used for OVCAR4 transduction. To mimic the spread of OC cells as clusters in ascites, $0.5 \times 10^6$ cells as preformed spheres in a mixture with $2.5 \times 10^6$ cells as single cell suspension in sterile saline were injected intraperitoneally (i.p.) into 6-week-old female ICR-SCID C.B-17 (IcrHan®Hsd-*Prkdc*$^{scid}$, Envigo) mice ($N = 10$). Tumor growth was followed by noninvasive bioluminescence imaging after i.p. injection of coelenterazine (35 µg in 100 µl PBS; Synchem) using Xenogen IVIS 100 imaging system (PerkinElmer). Carboplatin treatment (every 4 days, total of four doses; Accord) started on week 5

(25 mg/kg i.p. in sterile saline, $N = 5$). Control group ($N = 5$) received sterile saline injections. All mice received seven i.p. injections of an inert vehicle (30% PEG400 + 0.5% Tween80 + 5% Propylene glycol) used for an additional RSKi-treated group for which data are excluded from Results and Fig 3A due to extensive inhibitor liver toxicity. Mice were sacrificed on week 7.

In a separate experiment, the above model (CB17/IcrJcl-$Prkdc^{scid}$, CLEA) was used to evaluate MEKi. Carboplatin treatment (every 4 days, total of three doses; SelleckChem) started on day 25 (15 mg/kg i.p. in sterile saline, $N = 4$). Trametinib was additionally administered daily (1 mg/kg by oral administration in 4% DMSO + Corn oil, $N = 5$). Control group ($N = 4$) received injections of sterile saline and was also subjected to oral administration of 4% DMSO + Corn oil (this vehicle was also administered to carboplatin-treated mice). Mice were sacrificed on day 35.

A third independent proof-of-principle experiment was conducted (CB17/IcrJcl-$Prkdc^{scid}$, CLEA) to investigate the effects of RSKi BI-D1870 in a shorter time scale to avoid liver toxicity. Carboplatin treatment (every 4 days, total of two doses; SelleckChem) started on day 31 (15 mg/kg i.p. in sterile saline, $N = 4$). BI-D1870 was additionally administered on days 35–36 (25 mg/kg i.p. in 30% PEG400 + 0.5% Tween80 + 5% Propylene glycol, $N = 5$). Control group ($N = 4$) received injections of both vehicles. Mice were sacrificed on day 36, 5 h after the second injection of RSKi.

In all experiments, housing was in individually ventilated cages (IVC) with 4–5 mice per EU-standard sized cage with aspen-bedding. Nest boxes and material were provided as enrichments. Temperature was set to 20–24°C, relative humidity to 45–65%, and light rhythm to 12 h. Welfare was checked daily by the animal facility personnel and/or the study-conducting researchers.

## Immunofluorescence

For 2D immunofluorescence, cells grown on coverslips or in Nunc Lab-Tek Chamber Slides (Sigma) were fixed with 4% PFA for 15 min at RT, blocked with 5% BSA (Biowest) 0.1% Triton-X (Sigma) in PBS for 30 min at RT and incubated with primary antibody in 5% BSA in PBS for 1 h at RT. Cells were incubated with Alexa Fluor secondary antibodies (Thermo Scientific) in 5% BSA in PBS for 30 min at RT and mounted with VECTASHIELD Antifade Mounting Medium with DAPI (Vector Laboratories). To improve nuclear PAX8 staining, cells were post-fixed with ice-cold 1:1 mixture of acetone and methanol for 45 s before blocking.

For fixed 3D matrices, a 45 s post-fixation step with ice-cold 1:1 mixture of acetone and methanol was required. Matrices were then blocked with blocking buffer (15% FBS – 0.3% Triton-X in PBS) for 2 h at RT and incubated with primary antibody in blocking buffer overnight at 4°C. Multiple washing steps with 0.45% Triton-X in PBS were performed during the following day, and matrices were kept overnight at 4°C. Next, matrices were incubated with Alexa Fluor secondary antibodies and phalloidin in blocking buffer for 4 h at RT, washed several times with 0.45% Triton-X in PBS, and kept overnight at 4°C. After washing with PBS for a day, matrices were mounted on an object slide with VECTASHIELD Antifade Mounting Medium with DAPI.

OVCAR4 xenograft tumors as well as HGSC patient tumors embedded in Tissue-Tek OCT compound (Sakura Finetek) and frozen were cut into 10-μm cryosections and stained as follows.

Sections were melt in ice-cold PBS for 10 min, fixed in ice-cold acetone for 10 min, dried at RT for 10 min, washed with PBS, and blocked with TNB blocking buffer [0.1 M Tris–HCl, pH 7.5; 0.15 M NaCl; 0.5% (w/v) blocking reagent (PerkinElmer, Cat # FP1020)] for 30 min at RT. Primary antibody incubation was performed overnight at 4°C in a humidity chamber. After washes with TNT (0.1 M Tris-Cl, pH 7.5; 0.15 M NaCl; 0.1% (v/v) Tween 20), sections were incubated with Alexa Fluor secondary antibodies in TNB for 40 min, washed with TNT, and rinsed with PBS. Slides were mounted with VECTASHIELD Antifade Mounting Medium with DAPI.

Confocal micrographs of immunofluorescence stainings were obtained using a confocal microscope (LSM 780 and LSM 800) with a C-Apochromat 40×, 1.2 NA water objective lens and with a Plan-Apochromat 20×, 0.8 NA objective lens (all from Carl Zeiss).

## Tissue microarray

All patients diagnosed between 2002 and 2006 in Stockholm county with OC, fallopian tube, and primary peritoneal carcinoma and undesignated primary site according to the Swedish cancer registry were screened for eligibility. If not specified, all tumor types are herein referred as OC. Inclusion criteria were age above 18 years, high-grade serous histology, FIGO stages IIC to IV, no administration of chemotherapy prior to surgery or diagnostic biopsy, and availability of tissue from the tumor sites. Exclusion criteria were history of previous neoplastic disease (except for *in situ* cancer or basalioma), diagnosis at autopsy, prior chemotherapy, and not performed surgery or diagnostic biopsy. All cases were re-classified by a gynecological pathologist from the older three-tier differentiation grade to the new two-tier grade system (Malpica *et al*, 2004), and only high-grade serous tumors were selected.

Of the 401 screened for eligibility, 199 patients met the including criteria, of which 32 did not have available tissue and 31 missed histology data verification or clinical data; thus, 136 patients were included in the study (Fig 9A, Appendix Table S1). Clinical data were retrieved from the charts, coded, and collected in case report files. The FIGO stage was classified according to the 1988 system (Kandukuri & Rao, 2015). Response was defined according to RECIST and CA-125 criteria (Rustin *et al*, 2011).

Formalin-fixed and paraffin-embedded (FFPE) tumor tissues were retrieved from primary surgery or diagnostic biopsies derived from the chemo-naïve patients. A gynecological pathologist reviewed the tumor sections stained with hematoxylin and eosin, and representative areas of the tumor were chosen. Those were punched, and cores of 1 mm diameter were placed in a receiver TMA block. Two punches per patient were allowed, in line with the rules for TMA building at the Department of Pathology at Karolinska University Hospital; if possible, one punch was retrieved from the primary site and one from the metastatic omentum or peritoneum.

## Immunohistochemistry

Paraffin-embedded OVCAR4 xenograft tumors were cut into 7-μm-thick slides and stained with Click-iT™ TUNEL Colorimetric IHC Detection Kit (Thermo Scientific) and Ki67, following the manufacturer's instructions or as follows. Sections were deparaffinized and

rehydrated (3 × 5 min Tissue Clear, 3 × 2 min 99% ethanol, 2 × 2 min 95% ethanol, 1 × 2 min 70% ethanol, rinsed in water). Antigen retrieval was performed using 10 mM sodium citrate pH 6 (8-min strong heat, 20-min medium heat, 20-min cooldown). Endogenous peroxidase was quenched with 0.5% hydrogen peroxide for 30 min, 1 × 5 min water, and 1 × 5 min TBS. Sections were blocked with 2.5% normal horse serum for 30 min (ImmPRESS, Vector laboratories, Cat # MP-7402). Sections were incubated with Ki67 antibody diluted in 2.5% normal horse serum overnight at 4°C in a humidity chamber. Secondary antibody incubation was performed with ImmPRESS reagent (anti-mouse IgG coupled to peroxidase, ImmPRESS, Vector laboratories, Cat # MP-7402) for 30 min at RT. Staining was revealed using diaminobenzidine substrate (5-min incubation, 5 min in water). Sections were counterstained with aqueous hematoxylin (1-min incubation, rinsed with water), dehydrated (1 × 2 min 70% ethanol, 1 × 5 min 95% ethanol, 1 × 5 min 99% ethanol, 2 × 5 min Tissue Clear), and mounted with Pertex (Histolab, Cat # 00811).

Tissue microarrays blocks were cut into 4-µm-thick slides and stained for GPRC5A as described above with the following changes: Secondary antibody was anti-rabbit IgG coupled to peroxidase (ImmPRESS, Vector laboratories, Cat # MP-7401), incubation with diaminobenzidine substrate lasted 3 min.

Olympus IX73 microscope and 3DHISTECH Pannoramic 250 FLASH II digital slide scanner were used for imaging.

### TMA scoring

For the TMA scoring, a 4-point scale for GPRC5A-positive fraction (0: < 10% of area stained, 1: 10–50%, 2: 50–80%, 3: 80–100%) and maximum intensity of staining in cancer cells (0: negative; 1: mild; 2: moderate; and 3: strong) were used (Fig 9B; Liu *et al*, 2004; Thomson *et al*, 2009). Two metrics were produced: GPRC5A-positive fraction of total tumor tissue and maximum intensity of marker-positive area. For the survival analyses, GPRC5A maximum intensity was dichotomized in low (scores 0–2) and high (score 3). Twenty-five cores were reviewed separately by a second observer (Pinder *et al*, 2013) with a scoring concordance of 0.353 (kappa test; $P = 0.006$).

### Image analyses and statistics

Quantitative assessment of immunoblots was performed using Image Studio Lite, version 5.2. Quantification of IF signal intensity and positive IF and IHC were performed with ImageJ and QuPath (Bankhead *et al*, 2017).

All analyses were performed at least in triplicates, and the data distribution was tested by Kolmogorov–Smirnov and Shapiro–Wilk normality tests together with histogram analyses. Statistical significance was determined using two-sided Student's *t*-test and Mann–Whitney *U*-test. *P*-values are depicted as *$P < 0.05$, **$P < 0.01$, ***$P < 0.001$.

For animal studies, the number of mice per treatment group was limited to 4–6 to minimize the number of animals used yet provide adequate statistical power. Mice were randomly assigned into groups while simultaneously ensuring equal distribution of tumor burden (based on bioluminescent signal) within each group.

### The paper explained

**Problem**

Surgical removal of tumor masses and platinum chemotherapy are the standard treatment for HCSC, which is often widely spread at the time of diagnosis. Albeit initially effective in reducing tumor burden, the cycles of platinum treatment induce changes in the surgically inoperable treatment-escaping micro-metastases. These increasingly resistant, residual cells give rise to incurable, recurrent disease. This study assessed the long-neglected aspect of chemotherapy—the potentially oncogenic rewiring of cancer cell signaling induced by the platinum treatment *per se* as means to confer and sustain resistance.

**Results**

Using OC cell lines and patient-derived cultures, we have identified a platinum-induced, adaptive resistance mechanism involving EphA2 and RSK1/2 kinases and GPRC5A receptor. Inhibition of the oncogenic RSK-EphA2-pS897 signaling restored the tumor-suppressive EphA2-pY588 and specially sensitized HGSC cells with high GPRC5A expression to platinum *ex vivo*. Histological analysis of GPRC5A in a TMA with primary and metastatic HGSC specimens revealed its potential as a predictive marker for patient survival and treatment response.

**Impact**

The herein identified mechanism on how platinum chemotherapy induces an oncogenic RSK1/2-EphA2-GPRC5A signaling switch to sustain residual resistant cells reveals a targetable vulnerability to tackle them for complete tumor eradication. Importantly, this platinum-induced signaling axis entails also a potential prognostic marker for predicting survival and platinum treatment response: GPRC5A marker expression in HGSC could be used to stratify the unresponsive patients to combinatorial treatments targeting the oncogenic RSK-EphA2-pS897 axis.

For the analyses of the TMA, OS was defined as survival from date of diagnosis to date of death of any cause, in months. PFS was defined as the time-frame from the date of diagnosis to recurrence or death from any cause. ORR was defined as the proportion of patients with reduction in tumor burden at the end of treatment (EOT) measured according to the RECIST criteria: Cases with complete regression or partial regression (CR, PR) were considered responsive, while cases with stable disease or progressive disease (SD, PD) were considered non-responsive. Platinum sensitivity (including platinum alone or in combination with taxane) was defined according to the disease-free time after EOT: sensitive ($\geq$ 6 months disease-free), refractory (recurrence during treatment), or resistant (PD, recurrence < 6 months). In our analyses, refractory and resistant patients were pooled in one group. All the metrics relative to GPRC5A showing a significant *P*-value (< 0.05) at the univariate analysis (median age at diagnosis, type of surgery, residual tumor after surgery) as well as FIGO stage were entered into the multivariate model. Differences in OS and PFS, according to the clinico-pathological variables of interest, were estimated using logrank tests and Cox regression proportional hazard models. Correlation analyses were performed through Pearson chi-square test, gamma, and kappa tests. All TMA statistics were performed in SPSS, version 24.0.

For the analysis of independent publicly available HGSC cohorts, the TCGA dataset ($N = 578$, obtained through the TCGA data portal, http://cancergenome.nih.gov/) and the GSE49997 dataset ($N = 204$ (Pils *et al*, 2012); retrieved from the Gene Expression Omnibus

database; Edgar *et al*, 2002) were used. For TCGA dataset, cases that were healthy ($N = 8$), from different primary site than ovary/fallopian tube ($N = 4$), low-grade ($N = 87$), or with no information regarding histology or neoadjuvant chemotherapy ($N = 48$) were excluded from the analysis. Differences in OS in 40% highest vs. 40% lowest GPRC5A expressing patients were estimated using logrank test. This dataset was also used to investigate the correlation of EphA2 and GPRC5A mRNA with Spearman's rank test (cBioPortal for Cancer Genomics). For GSE49997 dataset, cases that were not of serous subtype ($N = 33$) or high-grade ($N = 46$) were excluded from the analysis. Differences in OS and PFS in patients with top 25% and low 25% EphA2, GPRC5A, RSK1-4 mRNA expression were estimated using logrank test. These statistics were performed in SPSS, version 26.0.

Expanded View for this article is available online.

## Acknowledgements

We thank M. Eriksson, L. Kogerman, and A. Tsuda for technical assistance; O. Carpen and S. Laín for cell lines; the Biomedicum Imaging Core at Karolinska Institutet for imaging facilities; the Histological Core Facility at Karolinska Institutet, the Histology Core at Karolinska Cancer Center, the Biomedicum Tissue Preparation and Genome Biology Units at University of Helsinki for histology facilities; and the Laboratory Animal Center at the University of Helsinki for animal husbandry. This work was funded by Karolinska Institutet, the KI Strategic Research Program in Cancer (StratCan-KICancer), the Swedish Cancer Society, the Swedish Research Council, the Cancer Research Foundations of Radiumhemmet, the Finnish Cancer Foundation, and the Doctoral Program in Integrative Life Science.

## Author contributions

Conception and design: LM-G, SPT, and KL. Acquisition of data (provided animals, acquired and managed patients, provided facilities, etc.): LM-G, EAP, SPT, SC, EH, DB, UJ, TA, YM, MY, AC, JJ, MS, HD, JWC, and KL. Analysis and interpretation of data (e.g., statistical analysis, biostatistics, computational analysis): LM-G, EAP, SPT, SC, DB, YM, JJ, and KL. Writing, review, and/or revision of the manuscript: LM-G, EAP, SPT, SC, EH, JJ, and KL. Study supervision: KL and SPT.

## Conflict of interest

The authors declare that they have no conflict of interest.

## For more information

(i)   Authors' webpage: https://ki.se/en/mtc/kaisa-lehti-group
(ii)  Gene Expression Omnibus: https://www.ncbi.nlm.nih.gov/geo/
(iii) The Cancer Genome Atlas: http://cancergenome.nih.gov/
(iv)  The Cancer Cell Line Encyclopedia: https://portals.broadinstitute.org/ccle

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
