## [Review Process File · EMBO Molecular Medicine]

Adaptive RSK-EphA2-GPRC5A signaling switch triggers chemotherapy resistance in ovarian cancer

Lidia Moyano-Galceran, Elina Pietilä, S. Pauliina Turunen, Sara Corvigno, Elisabet Hjerpe, Daria Bulanova, Ulrika Joneborg, Twana Alkasalias, Yuichiro Miki, Masakazu Yashiro, Anastasiya Chernenko, Joonas Jukonen, Madhurendra Singh, Hanna Dahlstrand, Joseph Carlson and Kaisa Lehti.

Review timeline:

Submission date:	19th July 2019
Editorial Decision:	16th September 2019
Revision received:	16th January 2020
Editorial Decision:	30th January 2020
Revision received:	5th February 2020
Accept:	6th February 2020

Editor: Jingyi Hou

Transaction Report:

1st Editorial Decision

16th September 2019

Thank you for the submission of your manuscript to EMBO Molecular Medicine. We have now received feedback from 2 out of 3 reviewers whom we asked to evaluate your manuscript. Given that both reviewers provide similar recommendations, we prefer to make a decision now in order to avoid further delay in the process.

As you will see from the reports below, the referees acknowledge the potential interest of the study. However, they also raise a number of concerns regarding your work, which should be convincingly addressed in a major revision of the present manuscript. In particular, it will be important to strengthen the underlying mechanism of RSK inhibition and to examine the effect of RSKi *in vivo* (as recommended by referee #2) to improve conclusiveness and clarity.

***** Reviewer's comments *****

Referee #1 (Comments on Novelty/Model System for Author):

1. A good effort has been made to use patient-derived cells to validate the findings and the original observation was made with some primary cultures of patient-derived cells. However, as the EphA2 pathway can impact on cell-cell contact and ECM adhesion and the authors studied HGSC cells and fibroblasts independently, results co-cultures would have been interesting.

Referee #1 (Remarks for Author):

Overall this is a well-performed study and the results are of translational interest. The authors have validated some of their work in primary patient-derived cultures which is commendable.

As the EphA2 pathways alter malignant cell interactions with their microenvironment, the paper would be strengthened by studying the effects of cisplatin and the inhibitors in 3D co-cultures of

malignant cells and fibroblasts even if they only use the cell lines.

In the Discussion the authors suggest that as RSKi have not yet entered clinical trial and may have off-target or toxic effects, MEK inhibitors may be useful clinically in HGSC in combination with platinum compounds. Investigation of this possibility would strengthen the translational impact of the paper.

Were all the cell lines used validated as HGSC origin?

Referee #2 (Remarks for Author):

In this article, Moyano-Galceran L et al. investigated the effect of targeting RSK to overcome resistance to platinum-based therapy in high-grade serous ovarian cancers (HG-SOC). The current work provides new insight indicating that chemotherapy-induced RSK-EphA2-GPRC5A signaling switch is associated to resistance to the platinum and that pharmacological RSK inhibition prevented platinum-induced oncogenic EphA2-S897 phosphorylation and EphA2-GPRC5A co-regulation, and in combination with platinum can counteract HG-SOC drug resistance. Further studies should offer more definite insight into the specific mechanisms through which RSK inhibition might represent a possible novel therapeutic strategy for HG-SOC patients. These findings need to be strengthened by additional experiments.

1. The hypothesis that platinum treatment leads to EphA2 upregulation and EphA2-pS897 phosphorylation in patient-derived HG-SOC cells *ex vivo* and in HG-SOC cell lines is intriguing, but the data are insufficient to offer more definite insight into the specific mechanism that regulate oncogenic EphA2 phosphorylation switch by platinum chemotherapy in HG-SOC cells.
2. In order to evaluate the effect of platinum in HG-SOC cells, cell vitality experiments should be corroborated by apoptotic analysis (Annexin V, Cleaved caspase, etc). Similarly, in order to evaluate whether combination treatment of platinum with RSKi could affect key mechanisms that are important for HG-SOC cell survival, the authors should analyze apoptosis.
3. The authors should provide a better understanding of the supportive role of RSK-EphA2-GPRC5A *in vivo*. How does this work? Is this via a decrease in apoptosis (TUNEL) or an increase in proliferation rate (Ki67)?
4. The inhibitor studies by using the pharmacological RSK inhibition to demonstrate the relevance of RSK-EphA2-GPRC5A signaling, should be complemented by the use of RNAi approaches (knock-down of EphA2 or GPCR5A or RSK).
5. To demonstrate that targeting RSK could improve platinum therapy response, the authors should analyze the effect of RSKi *in vivo*, in HG-SOC xenografts and PDX, providing indication whether combination with platinum can counteract HG-SOC drug resistance. Similarly, as suggested in point 2-3, the combination of RSKi and platinum should be evaluated in terms of apoptosis.
6. Moreover what happens in the experiment where "sensitive" HG-SOC cells are used and EphA2 or GPCR5A or RSK are overexpressed? Would these GPCR5A overexpressing cells now be less sensitive to platinum? This experiment might add further support for targeting this signaling pathway in HG-SOC.
7. The acquisition of drug resistance can be explained in part by intrinsic properties of cancer cells, but could be dependent also by the tumor microenvironment (TME). In order to demonstrate that targeting RSK can disable critical signaling in HG-SOC cells, but not alter cell vitality in cancer-associated fibroblasts (CAF), the authors should evaluate the heterogeneous expression of EphA2 or GPCR5A or RSK on tumor cells and CAF. Moreover, it should be interesting to evaluate host component effects in HG-SOC treated with RSKi in mono and combination therapy with platinum co-cultured in the absence and in the presence of CAF.
8. To better define GPCR5A as a predictive marker the authors should provide more details using larger cohort of platinum-sensitive and resistant HG-SOC patients. Moreover the authors should

evaluate the expression of RSK-EphA2-GPRC5A axis as predictor signature of poor platinum-based therapy responses and shorter survival in HG-SOC patients.

1st Revision - authors' response

16th January 2020

***** Reviewer's comments *****

Referee #1 (Comments on Novelty/Model System for Author):

A good effort has been made to use patient-derived cells to validate the findings and the original observation was made with some primary cultures of patient-derived cells. However, as the EphA2 pathway can impact on cell-cell contact and ECM adhesion and the authors studied HGSC cells and fibroblasts independently, results co-cultures would have been interesting.

Response: We thank Reviewer #1 for the constructive comments and agree that co-cultures of HGSC cells and fibroblasts are relevant and of interest to test the impact of tumor microenvironment on the identified RSK-EphA2-GPRC5A signaling axis. To address this point, we have now established 3D co-culture models using OC patient-derived cancer associated fibroblasts (CAFs), and included the results from RSKi and platinum treatments of these co-culture experiments in the new **Fig 7** and **Appendix Fig S6** of revised manuscript. Please see more detailed description of the co-culture results below in the response to specific comment 1.

Referee #1 (Remarks for Author):

1. Overall this is a well-performed study and the results are of translational interest. The authors have validated some of their work in primary patient-derived cultures which is commendable. As the EphA2 pathways alter malignant cell interactions with their microenvironment, the paper would be strengthened by studying the effects of cisplatin and the inhibitors in 3D co-cultures of malignant cells and fibroblasts even if they only use the cell lines.

Response: We thank the reviewer for interest in our results and for this highly relevant suggestion to use co-cultures of cancer cells and fibroblasts. To study the effects of cisplatin and RSKi in OC-CAF co-culture, we first generated red fluorescent OVCAR8-RFP cells, and allowed them to form spheroids as mono- or co-cultures with green fluorescent BjhTERT-GFP fibroblast cell line in non-adherent conditions, followed by treatment with cisplatin and RSKi alone as well as in combination. Using RFP signal intensity at the starting point as well as after 48h and 72h of treatments as a measure of OC cell content, we validated the activity of RSKi, cisplatin and their combination in the 3D cell spheroids. In OVCAR8-RFP monocultures, single treatments with RSKi or cisplatin inhibited RFP signal relative to

untreated control, but most notably, combination treatment with RSKi and cisplatin reduced OC cells (RFP intensity) even relative to the treatment start, suggestive of effective cell killing. The combination treatment reduced OVCAR8-RFP most effectively also in co-culture with BjhTERT-GFP fibroblasts. Please see these results below in Figure 1 for review.

This data, although further supporting the efficacy of cisplatin-RSKi combination, did not allow us to separate apoptosis and proliferation. Moreover, the strong induction of OVCAR8 cell content by BjhTERT fibroblasts compared to monocultures appeared to contrast with the effect of OC patient-derived CAFs on OVCAR8 in our following experiments now included in the revised manuscript (see below). Therefore, we decided not to include this BjhTERT-OVCAR8 co-culture data in the manuscript. It is included here, however, to further support the suggestion that the cisplatin-RSKi combination can sensitize the cells to platinum-based therapy, as well as to show how differential impact different fibroblasts can have on the OC cells.

Figure 1 for Review. Chart illustrates the relative changes in OVCAR8-RFP fluorescence signal over time in spheroid mono- and co-cultures with BjhTERT fibroblasts treated with cisplatin and Ljh685 alone or as a combinatorial treatment (48 h and 72 h, N = 3).

As suggested by both Reviewer#1 and #2, to analyze in a highly relevant 3D microenvironment the specific effects of platinum and RSKi treatments in OC apoptosis and proliferation, we prepared spheroids of OVCAR8-RFP and OC patient-derived CAFs under non-adherent conditions, embedded them in 3D collagen and allowed the cells to grow for 4 days. After subsequent treatments with cisplatin and RSKi alone and in combination for 20 h, apoptosis was assessed by immunofluorescence for cleaved caspase-3 (clCasp3). In OVCAR8-RFP 3D monocultures clCasp3/apoptosis was increased by cisplatin alone and by combining Ljh685 with cisplatin (new **Fig 7A-B**).

In patient CAF monocultures, cisplatin alone or in combination with Ljh685 did not affect clCasp3/apoptosis. Notably, in the 3D co-culture with CAFs, cisplatin or Ljh685 alone also failed to enhance OVCAR8-RFP apoptosis significantly, whereas the combined Ljh685-cisplatin treatment increased clCasp3 in OVCAR8-RFP (new **Fig 7A-B**; $p = 0.022$).

To validate these findings in an even more clinically relevant model, we generated co-cultures of patient-derived OCKI_p13 HGSC cells and OCKI_p22 CAFs and treated them as described above. Notably, combination of LJH685 with cisplatin further enhanced the platinum-induced cancer cell apoptosis (clCasp3) in the co-culture ($p = 0.002$). These results, indicating that in the relevant TME, RSKi-cisplatin combination can sensitize OC cells to platinum-based therapy, have now been included in the new **Fig 7A-D**, **Appendix Fig S6A-B** and described in Results on p. 12 of the revised manuscript.

2. In the Discussion the authors suggest that as RSKi have not yet entered clinical trial and may have off-target or toxic effects, MEK inhibitors may be useful clinically in HGSC in combination with platinum compounds. Investigation of this possibility would strengthen the translational impact of the paper.

Response: We appreciate the importance of providing experimental evidence to discuss and suggest such translational implications. In the originally submitted manuscript, we had only used MEKi UO126 (at a concentration of 10 μ M), showing efficient inhibition of the cisplatin-induced EphA2-pS897 in TYK-nu cells (Figure 4E). To address this relevant point, we first investigated the *in vitro* efficacy of two additional MEKi, Refametinib and Trametinib, the latter being an FDA approved drug for solid tumors (1, 2). Both these MEKi showed good inhibition of the EphA2 oncogenic EphA2-pS897, while UO126 (now used at a lower concentration of 1 μ M based on literature search) was inefficient, and thus not used for further experiments.

Please see new **Appendix Fig S3E-F** and Results on p. 10: “Coincident with ERK1/2-pT202/Y204 inhibition, Trametinib decreased viable OVCAR4 by over 40% compared to untreated control, whereas the relative cell cisplatin sensitivity remained unaltered”. To further address this point, we examined the effects of Trametinib-carboplatin combination treatment in cell death (TUNEL) and proliferation (Ki67) in the model of OVCAR4 xenografts *in vivo*. Trametinib in combination with carboplatin did not induce more apoptosis or significantly reduce proliferation than carboplatin treatment alone.

These results have now been included in **Appendix Fig S3G-J** and described in the revised manuscript Results on p. 10. Altogether, this data suggests that the inhibition of broadly acting, proliferation-driving MEK-ERK1/2 pathway has a different mode of action that may not be directly relevant and related to the specific inhibition of the RSK-EphA2-GPRC5A axis affected by combinatorial treatment with RSKi and platinum. To address this consideration, we have now revised this aspect in Discussion in p. 17.

3. Were all the cell lines used validated as HGSC origin?

Response: This is an important question, since recent studies have brought into light striking findings on the use of cells that do not recapitulate the mutational landscape of specific OC subtypes, such is the case of HGSC: up to 90% of published studies using “HGSC” cell lines are actually based on cells that are TP53 wild-type and instead have characteristic mutations of other OC subtypes (3).

In this study we used TP53 mutant OVCAR3, OVCAR4, OVCAR8, TYK-nu and TYK-nu.R cell lines, from which OVCAR3, OVCAR4, TYK-nu and TYK-nu.R have been ranked based on the genetic alterations as highly likely HGSC, while OVCAR8 cell line has been described as likely HGSC (4). This reference and information have now been included in the cell line description in Materials and Methods on p. 19 of the revised manuscript. The HGSC patient-derived passaged cells OCKI_p01 - OCKI_p11 used in the experiments have also been analyzed by TP53 sequencing and tested for nutlin insensitivity as an indication of TP53 mutation status as shown in Appendix Fig S2C. The description of patient-derived cells has also been revised in Materials and Methods on p. 19-20 to clarify the presentation.

Referee #2 (Remarks for Author):

In this article, Moyano-Galceran L et al. investigated the effect of targeting RSK to overcome resistance to platinum-based therapy in high-grade serous ovarian cancers (HG-SOC). The current work provides new insight indicating that chemotherapy-induced RSK-EphA2-GPRC5A signaling switch is associated to resistance to the platinum and that pharmacological RSK inhibition prevented platinum-induced oncogenic EphA2-S897 phosphorylation and EphA2-GPRC5A co-regulation, and in combination with platinum can counteract HG-SOC drug resistance. Further studies should offer more definite insight into the specific mechanisms through which RSK inhibition might represent a possible novel therapeutic strategy for HG-SOC patients. These findings need to be strengthened by additional experiments.

Response: We thank Reviewer #2 for the interest in our manuscript and for the constructively critical comments that helped us to improve our manuscript. To better define the specific mechanism and strengthen the manuscript, we have performed various additional experiments during the revision, included the results and addressed the points as described below.

1. The hypothesis that platinum treatment leads to EphA2 upregulation and EphA2-pS897 phosphorylation in patient-derived HG-SOC cells *ex vivo* and in HG-SOC cell lines is intriguing, but the data are insufficient to offer more definite insight into the specific mechanism that regulate oncogenic EphA2 phosphorylation switch by platinum chemotherapy in HG-SOC cells.

Response: We thank the Reviewer for this constructive criticism, which motivated us to conduct several new experiments and allowed important improvements to the manuscript in terms of the specific mechanism underlying the oncogenic EphA2 phosphorylation switch linked to chemoresistance in HGSC cells.

The following specific insights on these mechanisms have now been included in the revised version of our manuscript:

A) Clarified presentation of our original results showing that platinum induces ERK1/2-RSK pathway activation, which correlates with the EphA2-S897 phosphorylation (Figure 4H-I in the original and revised manuscript).

B) New results of siRNA experiments demonstrating that the depletion of specific activity of RSK2 in OVCAR8/4, as well as of RSK1 or RSK2 in TYK-nu and TYK-nu.R, will lead to the tumor suppressive EphA2 serine-to-tyrosine reversal (new **Fig 6D-F**; new **Fig EV3A-D** and described in the revised manuscript Results on p. 11-12). Silencing RSK2 specifically restored EphA2-pY588 in OVCAR4 and OVCAR8. This result indicates the

essential functions of these two RSKs in the platinum-induced oncogenic EphA2 switch.

C) New results of RSK1 and RSK2 siRNA experiments revealing an essential RSK1 function in the platinum-induced GPRC5A regulation in both the resistant OVCAR8 and TYK-nu.R, in association with apoptosis evasion, as indicated by PARP cleavage as well as reduced survival after RSK1 knockdown (new **Fig 6E-F**; new **Fig EV3**).

D) New results highlighting exclusive GPRC5A induction in HGSC cells, but not in corresponding tumor stroma or CAFs (new **Fig 8A-B**; **Appendix Fig S7A**).

The results C) and D) combined, i.e. the specific function of RSK1 in GPRC5A regulation coupled to cancer-specific GPRC5A induction, will help to explain the specific sensitization of the malignant cells to platinum-induced apoptosis, while stroma remains protected.

Platinum chemotherapy is known to induce oxidative stress/ROS-related ERK1/2 activation in different types of malignant and non-malignant cells (5). This notion has now also been included in the revised Discussion. In the originally submitted manuscript, we showed that both ERK1/2 and RSK were activated upon platinum treatment in our OC cell models (Figure 4H and I). Further, we showed that the EphA2-pS897 phosphorylation was repressed by pharmacological inhibition of either MEK-ERK1/2 axis or its downstream target RSK (RSK1-3 members/proteins inhibited by both the inhibitors; Figure 4E-I).

To clarify specifically the mechanism of the broadly acting ERK1/2-RSK axis in the EphA2 phosphorylation switch, we have now additionally assessed the effects of MEKi in OC platinum responses (new **Appendix Fig 3E-J**, see also response to comment 2 of Reviewer #1) and RSK expression of RSK1-4 in the OC cells used in the experiments (new **Fig EV3A**).

2. In order to evaluate the effect of platinum in HG-SOC cells, cell vitality experiments should be corroborated by apoptosis analysis (Annexin V, Cleaved caspase, etc). Similarly, in order to evaluate whether combination treatment of platinum with RSKi could affect key mechanisms that are important for HG-SOC cell survival, the authors should analyze apoptosis.

Response: We thank the reviewer for this excellent suggestion.

To evaluate the effect of platinum in HGSC cells, we assessed apoptosis (cleaved caspase-3; cCasp3) in patient-derived cancer cells cultured in 3D collagen. In this setting, cisplatin treatment significantly increased apoptosis ($p = 0.022$; See new **Fig EV1D-E**).

Further, to assess the effects of platinum alone or combined with RSKi on apoptosis in another highly relevant cell culture model, we stained for cCasp3 3D collagen OVCAR8-RFP spheroids treated with cisplatin alone or in combination with LJH685. When used alone, cisplatin treatment

induced apoptosis, while LJH685 alone had no significant effect on apoptosis. When used in combination, LJH685-cisplatin significantly induced apoptosis/cIcasp3 ($p = 0.002$). Likewise, in a 3D co-culture model of patient-derived HGSC cells and CAFs, cisplatin- LJH685 combination further enhanced platinum-induced apoptosis. These results have now been included in new **Fig 7A-D, Appendix Fig S6A-B** and described in the revised manuscript Results p. 12.

To directly associate the RSK1/2 activities and EphA2-GPRC5A co-regulation to apoptosis and proliferation, we further detected cleaved PARP as a marker of apoptosis along with the proliferation marker PCNA by immunoblotting in RSK1/2 siRNA transfected TYK-nu.R. In the platinum-treated cells, siRNA-mediated RSK1 depletion led to GPRC5A suppression coincident with increased cleaved PARP (**Fig 6E**). Instead, RSK2 depletion increased cleaved PARP in the absence of cisplatin, whereas the proliferation marker PCNA was generally less affected by RSK1/2 knockdown, and even increased after cisplatin treatment in the resistant cells (**Fig 6E**).

Altogether, these results indicate that both platinum and RSKi-platinum combination treatments primarily decrease cell viability via increased apoptosis.

3. The authors should provide a better understanding of the supportive role of RSK-EphA2-GPRC5A *in vivo*. How does this work? Is this via a decrease in apoptosis (TUNEL) or an increase in proliferation rate (Ki67)?

Response: This is a valid point. To further address the role of RSK-EphA2-GPRC5A *in vivo*, we analyzed the OVCAR4 xenograft tumors presented in the originally submitted manuscript for EphA2, EphA2-pS897, GPRC5A, cleaved caspase-3 (cIcasp3), TUNEL and Ki67. Carboplatin treatment increased EphA2, EphA2-pS897, cIcasp3 and TUNEL, but did not alter Ki67. Notably, EphA2-pS897 and cIcasp3 localized to different tumor cells and areas in the carboplatin treated tumors. Therefore, we conclude that the treatment-escaping HGSC cells activated oncogenic EphA2 signaling to evade apoptosis in response to platinum chemotherapy *in vivo*. These results have now been included in **Fig 3E-I, Fig EV2B-D** and described in Results p. 8.

Further, in two independent *in vivo* experiments, cisplatin treatment likewise induced apoptosis (detected by TUNEL) but had no major effect on proliferation (assessed by Ki67). In one of these experiments, BI-D1870 was used in combination with platinum for 48 h, significantly increasing TUNEL/apoptosis (2.5 ± 1.8 fold, $p = 0.029$) but not affecting Ki67/proliferation. These results (presented in **Fig 7E-H, Appendix Fig 3G-J**) suggest that both platinum and RSKi-platinum combination treatments decreased OC cell viability via increased apoptosis.

4. The inhibitor studies by using the pharmacological RSK inhibition to demonstrate the relevance of RSK-EphA2-GPRC5A signaling, should be complemented by the use of RNAi approaches (knock-down of EphA2 or GPCR5A or RSK).

Response: We thank the reviewer for this relevant suggestion. To further elucidate the RSK-EphA2-GPRC5A axis, we silenced RSK1 and/or RSK2 (the cancer-associated RSK proteins highly expressed in OC cells; see new Fig EV3A) in OC cells. As mentioned above, RSK1 and/or RSK2 depletion prevented the platinum-induced EphA2 phosphorylation switch in all OC cells. In OVCAR4 and OVCAR8, RSK2 silencing specifically restored EphA2-pY588, whereas RSK1 knockdown in the platinum resistant TYK-nu.R and OVCAR8 inhibited the increase of GPRC5A 46 KDa form and increased apoptosis as reflected by cleaved PARP (**Fig 6E**). These results have now been included in **Fig 6D-E**, the new **Fig EV3A-B** and described in the revised manuscript Results on p. 11-12.

In the originally submitted manuscript we had performed siRNA-mediated silencing of EphA2 and GPRC5A (see Appendix Figure S3C-D and S5A-C). We showed minor effect of siEphA2 on cell viability, which was consistent with our conclusion “Rather than blocking the entire signaling duality by EphA2 knockdown, the specific RSK-EphA2-pS897 inhibition and reversal to tumor-suppressive EphA2-pY588 correlated with the effective OC cell sensitization to platinum”. We also noted an induction of GPRC5A upon siEphA2 as well as induction of EphA2 (total and pS897) upon siGPRC5A in TYK-nu cells. We have now clarified these results in manuscript p. 11.

5. To demonstrate that targeting RSK could improve platinum therapy response, the authors should analyze the effect of RSKi *in vivo*, in HG-SOC xenografts and PDX, providing indication whether combination with platinum can counteract HG-SOC drug resistance. Similarly, as suggested in point 2-3, the combination of RSKi and platinum should be evaluated in terms of apoptosis.

Response: To address this question, we conducted pilot experiments to test BI-D1870 in combination with carboplatin in two experimental set-ups with the same dosing (25 mg/kg *i.p.*): every two days for two weeks, or daily for 2 days. However, the longer dosing scheme of BI-D1870 in combination

Figure 2 for Review. Manifestation of severe liver toxicity with BI-D1870 in combination with carboplatin. A) Plasma samples of female SCID mice 14 days after the treatment start. Controls (black), carboplatin treated (orange), and carboplatin + 25 mg/kg BI-D1870 for seven times every two days (green, outlined). Note the yellowish shade of plasma (*) as a result of liver failure in 3/6 mice in this group. B) Icterus (#) of a mouse in the carboplatin + BI-D1870 group at sacrifice.

with carboplatin resulted in severe liver toxicity, icterus, weight loss and increased red blood cell sedimentation rate in 50% of the treated mice (see below in Figure 2 for Review) forcing us to quit the dosing. From the shorter dosing scheme, we analyzed apoptosis by TUNEL staining, which revealed a significant induction of apoptosis compared to the control group ($p = 0.029$). We think that these results serve as convincing proof-of-principle for the mode of action by apoptosis induction. They have now been included in **Fig 7E-H** and described in the revised manuscript Results on p. 13.

To be successful in the *in vivo* tumor models, the inhibitor molecules need to entail well-optimized pharmacokinetic properties. To best of our knowledge, no good RSKi with favorable pharmaco-kinetic/-dynamic (PK/PD) characteristics *in vivo* has yet been developed. The commonly used *in vitro* inhibitors of RSK, LJM685 and BI-D1870, have both been preliminarily tested in PK/PD studies, showing poor drug stability, high clearance and short plasma half-life (6-9). Although the poor PK of RSKi could be overcome via comprehensive compound optimization, it is far beyond the scope of this study. However, as mentioned above, we conducted small RSKi pilot experiments in our orthotopic model of metastasized ovarian cancer in female SCID mice. The number of mice in these experimental groups was limited to 4-6, as we firmly think that the ethical 3R (replace, reduce, refine) principle governing all animal work should be the principal guideline when working with compounds with known suboptimal PK.

This 3R principle also guided our decision to not generate the PDX models suggested by the reviewer for treatment with these compounds. Moreover, the extensive efforts and time frames required for a PDX study would be out of the scope for the revision of this study and will, in our opinion, require another independent study.

6. Moreover what happens in the experiment where "sensitive" HG-SOC cells are used and EphA2 or GPCR5A or RSK are overexpressed? Would these GPCR5A overexpressing cells now be less sensitive to platinum? This experiment might add further support for targeting this signaling pathway in HG-SOC.

Response: To address this relevant point, we overexpressed RSK1/2, GPCR5A or EphA2 in cisplatin sensitive OVCAR4 and also in treatment-resistant OVCAR8 (only RSK1/2). Upon cisplatin treatment, the viability of the platinum sensitive OVCAR4 cells overexpressing RSK1/2 increased (5 μ M cisplatin: RSK1_OE 106.0 ± 2.3 % and RSK2_OE 107.6 ± 6.1 % vs control 89.9 ± 4.9 %, $p < 0.048$). This increase in viability upon treatment was also seen in the more resistant OVCAR8 overexpressing RSK1. In OVCAR4 overexpressing GPCR5A or EphA2, viability was also increased when compared to treated control (GPCR5A OE at 10 μ M cisplatin: $41.3 \pm$

0.3 % vs 2.5 ± 24.7 % in control, $p < 0.002$; increased viability after EphA2 OE at 5 μ M cisplatin: 100.7 ± 0.3 % vs 84.6 ± 10.4 % in control, $p < 0.029$). These results have now been included in the new **Fig EV3C-F** and in Results p. 12.

7. The acquisition of drug resistance can be explained in part by intrinsic properties of cancer cells, but could be dependent also by the tumor microenvironment (TME). In order to demonstrate that targeting RSK can disable critical signaling in HGSC cells, but not alter cell vitality in cancer-associated fibroblasts (CAF), the authors should evaluate the heterogeneous expression of EphA2 or GPRC5A or RSK on tumor cells and CAF. Moreover, it should be interesting to evaluate host component effects in HG-SOC treated with RSKi in mono and combination therapy with platinum co-cultured in the absence and in the presence of CAF.

Response: To address this interesting suggestion by the reviewer, we have now assessed the expression of RSK, EphA2 and GPRC5A in HGSC frozen tissue sections by immunofluorescence and in patient-derived HGSC cells and CAFs by immunoblotting. Immunofluorescence showed that RSK and EphA2 expression was higher in the cancer cells than in the stroma and that GPRC5A localized exclusively in the areas with cancer cells. Immunoblotting for RSK, EphA2 and GPRC5A also showed that despite variable expression of these proteins in different patient-derived cancer cells, they were notably more expressed in cancer cells than in CAFs. These results have now been included in **Fig 8A-B, Appendix Fig S7A** and described in the revised manuscript Results on p. 13.

To assess the host component effect in HGSC cells apoptosis, we generated mono- and co-culture spheroids of OVCAR8-RFP/OCKI_p13 cancer cells and patient-derived CAFs and embedded them in 3D collagen. Please see the answer to Reviewer #1, question 1 for detailed description of these results.

We think that these results altogether have markedly improved the revised manuscript. Firstly, the cancer-specific GPRC5A induction, coupled with the direct link between RSK function and platinum induced GPRC5A regulation identified during revision when performing the siRNA experiments (**Fig 6, Fig EV3B**), provide a more specific mechanistic insight into the revised study. Secondly, our OC-CAF co-culture results highlight that even in a culture where CAFs seem to further inhibit OVCAR8 platinum response, the RSKi-cisplatin combination can induce apoptosis (new **Fig 7**).

8. To better define GPRC5A as a predictive marker the authors should provide more details using larger cohort of platinum-sensitive and resistant HG-SOC patients. Moreover the authors should evaluate the expression of RSK-EphA2-GPRC5A axis as predictor signature of poor platinum-based therapy responses and shorter survival in HG-SOC patients.

Response: We have now clarified the definition of platinum sensitive and resistant HGSC patients (N=136) according to clinical standards regarding the treatment (one should not consider platinum as single agent when defining patient groups, but rather group together patients with platinum single and double treatments). We have accordingly modified the Appendix Materials and Methods section in p. 17, **Fig 9E** and **Appendix Table S7** as well as the Results in p. 14.

Moreover, we have analyzed two independent HGSC cohorts (TCGA dataset for OC with N = 578, <http://cancergenome.nih.gov/>; GSE4997 dataset with N = 204, (10)) to validate the power of the RSK-EphA2-GPRC5A signaling axis as predictor signature. Survival analysis of the TCGA dataset validated our findings on GPRC5A association with worse overall survival in this case at the mRNA level. Moreover, survival analysis of the GSE4997 cohort further uncovered the potential of the combination of EphA2+GPRC5A mRNA expression as an approach to predict progression-free survival of the patients ($p = 0.020$ for EphA2+GPRC5A with and without RSK1 and RSK2). These results have now been included in **Fig 9F-H**, **Fig EV5** and described in the revised manuscript Results on p. 14-15.

References

1. Odogwu L, Mathieu L, Blumenthal G, Larkins E, Goldberg KB, Griffin N, et al. FDA Approval Summary: Dabrafenib and Trametinib for the Treatment of Metastatic Non-Small Cell Lung Cancers Harboring BRAF V600E Mutations. *Oncologist*. 2018;23(6):740–5.
2. Wright CJ, McCormack PL. Trametinib: first global approval. *Drugs*. 2013;73(11):1245–54.
3. Beaufort CM, Helmijr JC, Piskorz AM, Hoogstraat M, Ruigrok-Ritstier K, Besselink N, et al. Ovarian cancer cell line panel (OCCP): clinical importance of in vitro morphological subtypes. *PLoS One*. 2014;9(9):e103988.
4. Domcke S, Sinha R, Levine DA, Sander C, Schultz N. Evaluating cell lines as tumour models by comparison of genomic profiles. *Nat Commun*. 2013;4:2126.
5. Dasari S, Bernard Tchounwou P. Cisplatin in cancer therapy: Molecular mechanisms of action. *European Journal of Pharmacology*. 2014;740:364–78.
6. Casalvieri KA, Matheson CJ, Backos DS, Reigan P. Selective Targeting of RSK Isoforms in Cancer. *Trends Cancer*. 2017;3(4):302–12.
7. Jain R, Mathur M, Lan J, Costales A, Atallah G, Ramurthy S, et al. Discovery of Potent and Selective RSK Inhibitors as Biological Probes. *J Med Chem*. 2015;58(17):6766–83.
8. Hammoud L, Adams JR, Loch AJ, Marcellus RC, Uehling DE, Aman A, et al. Identification of RSK and TTK as Modulators of Blood Vessel Morphogenesis Using an Embryonic Stem Cell-Based Vascular Differentiation Assay. *Stem Cell Reports*. 2016;7(4):787–801.

9. Pambid MR, Berns R, Adomat HH, Hu K, Triscott J, Maurer N, et al. Overcoming resistance to Sonic Hedgehog inhibition by targeting p90 ribosomal S6 kinase in pediatric medulloblastoma. *Pediatr Blood Cancer*. 2014;61(1):107–15.
10. Pils D, Hager G, Tong D, Aust S, Heinze G, Kohl M, et al. Validating the impact of a molecular subtype in ovarian cancer on outcomes: a study of the OVCAD Consortium. *Cancer Sci*. 2012;103(7):1334–41.

2nd Editorial Decision

30th January 2020

Thank you for the submission of your revised manuscript to EMBO Molecular Medicine. We have now received the enclosed report from the reviewer who was asked to re-assess it. As you will see the reviewer is now overall supportive and I am pleased to inform you that we will be able to accept your manuscript pending the following amendments:

***** Reviewer's comments *****

Referee #2 (Remarks for Author):

The authors have presented a novel approach with mechanistic underpinnings of how to improve currently available platinum-based therapy for high-grade serous ovarian cancers. The authors have responded to all of the reviewers' comments including providing considerable additional data. I have no further concerns.

Corresponding Author Name: Dr. Kaisa Lehti
Journal Submitted to: EMBO molecular medicine
Manuscript Number: EMM-2019-11177